# Enhancing Conformal Prediction via Class Similarity

Ariel Fargion [1]   Lahav Dabah [1]   Tom Tirer [1]

## Abstract

Conformal Prediction (CP) has emerged as a powerful statistical framework for reliable classification, which generates a prediction set, guaranteed to include the true label with a pre-specified probability. The performance of CP methods is typically assessed by their average prediction set size. In setups where the classes can be partitioned into semantic groups, e.g., based on shared downstream actions or more interpretable coarse labels, users can benefit from prediction sets that are not only small but also contain a limited number of groups. This paper begins by addressing this problem and ultimately offers a widely applicable tool for boosting any CP method on any dataset. First, given a class partition, we propose augmenting the CP score function with a term that penalizes predictions with "out-of-group" errors. We theoretically analyze this strategy and prove its advantages for group-related metrics. Surprisingly, we show mathematically that, for common class partitions, it can also reduce the average set size of *any* CP score function. Our analysis reveals the class-similarity factors behind this improvement and motivates a variant that can further reduce prediction set size by leveraging the model's embeddings, *without requiring any human semantic partition*. Finally, we present an extensive empirical study, encompassing prominent CP methods, multiple models, and several datasets, which demonstrates that our class-similarity-based approach consistently enhances CP methods.

## 1. Introduction

Conformal Prediction (CP) (Vovk et al., 1999; 2005) has emerged as a powerful statistical framework for high-stakes classification applications, such as medical diagnoses (Lambert et al., 2024) and autonomous vehicle decision-making (Lindemann et al., 2024). Rather than predicting a single label, the CP framework outputs a set of candidate labels with a formal guarantee of marginal coverage: under exchangeability of the calibration and test samples, the prediction set will include the correct label with a user-specified probability. This property makes CP particularly valuable in safety-critical domains, where missing the correct label can have severe consequences. A key metric for comparing different CP methods is the average size of their prediction sets, commonly referred to in the literature as *efficiency* (Sadinle et al., 2019; Angelopoulos et al., 2021; Huang et al., 2024; Dabah and Tirer, 2025).

In many practical scenarios, classes can be grouped into semantic categories ("superclasses"), e.g., based on shared downstream actions or more interpretable coarse labels. In settings where such semantic groupings align with decision-making, users can benefit from prediction sets that are not only small but also semantically aligned. For instance, in disease recognition, a prediction set consisting of conditions that require similar treatment can help clarify appropriate next steps. Similarly, in species classification, confusing two species within the same family (e.g., sparrow species) is often less consequential for a typical user than confusing species from different families (e.g., sparrow versus hawk). Accordingly, exploiting such class-level structure, while preserving standard conformal coverage guarantees and without degrading efficiency, can be beneficial.

However, while current CP approaches ensure that the true label is included in the prediction set with the pre-specified probability, they typically do not account for the semantic coherence of labels within the prediction set, and recent works that incorporate label structure (Goren et al., 2024; Zhang et al., 2025; Hengst et al., 2025) lead to prediction sets that contain substantially more classes than baseline methods.

This paper addresses this gap and extends beyond it, ultimately offering a general tool for improving the efficiency of any CP method on any dataset. First, assuming a known partition of classes into groups, we propose augmenting the CP score function with a regularization term that penalizes predictions with "out-of-group" errors. We provide a theoretical analysis of this strategy and prove that it reduces the

---

[1]Faculty of Engineering, Bar-Ilan University, Ramat Gan, Israel. Correspondence to: Ariel Fargion <arielfar77@gmail.com>.

*Proceedings of the 43$^{rd}$ International Conference on Machine Learning*, Seoul, South Korea. PMLR 306, 2026. 

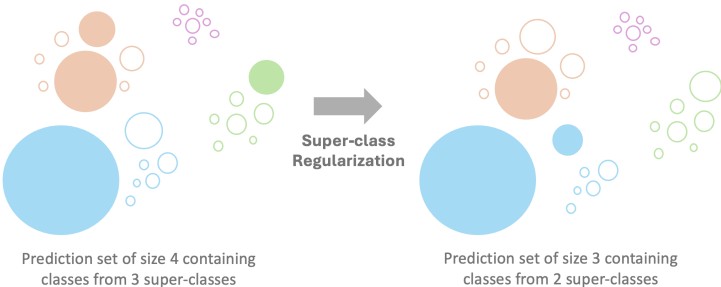

Figure 1. Illustration of prediction sets for an example before and after applying our proposed regularization. Each circle corresponds to a class, with colors indicating superclasses. Filled circles denote classes included in the prediction set, and circle size reflects the softmax value. In this example, the prediction set size decreases from 4 to 3, and the number of superclasses represented decreases from 3 to 2. We show that our regularization reduces the average prediction set size in typical setups, regardless of the baseline score function.

expected number of unique groups appearing in a prediction set. Interestingly, our theory also reveals a surprising property: for common class partitions, applying this penalty can simultaneously decrease the average prediction set size, regardless of the underlying CP score function. Our analysis further identifies the class similarity factors behind this improvement.

Motivated by these insights, we extend our approach and propose a model-specific variant, *which does not require any human semantic partition*. Specifically, we construct a class similarity matrix from the classifier's embedding vectors, leveraging the model's own perception of similar classes. This enables regularization that further reduces the prediction set size and does not require any external knowledge of class groups, making it applicable for general datasets.

Finally, we present an extensive empirical study evaluating the performance of both variants, the one that uses a known, Model-Agnostic (MA), class partition, and the one that relies on Model-Specific (MS) class similarities. Our experiments encompass multiple datasets, models, and prominent CP score functions: LAC (Sadinle et al., 2019), RAPS (Angelopoulos et al., 2021), and SAPS (Huang et al., 2024). We show that our class-similarity-based approach consistently enhances each of these diverse CP methods, providing a flexible and widely applicable tool for improving both the coherence and efficiency of prediction sets. To the best of our knowledge, no previous post-training approach has consistently outperformed the standard LAC in terms of average prediction set size.

**Summary of our main contributions.**

- We propose an "out-of-group" penalty approach, independent of the original CP score function, which improves both the semantic coherence of prediction sets and their average size.
- We provide a theoretical analysis of the proposed penalty, proving not only its effectiveness in reducing the expected

number of unique groups in the prediction set but also its non-intuitive ability to decrease the average set size for any score function.

- We introduce a model-specific variant for the approach, based on the model's internal embeddings, which further reduces the prediction set size and does not require any known class structure.
- We conduct an extensive empirical evaluation across multiple dataset-model pairs, demonstrating that both our model-agnostic and model-specific variants consistently enhance prominent CP methods, outperforming their original versions in both semantic coherence and size of their prediction sets.

## 2. Related Work

**Clustered and group-conditional coverage CP.** Several works (Vovk, 2012; Ding et al., 2023; Bairaktari et al., 2025) aim to improve coverage across heterogeneous label groups. These methods typically apply the CP procedure separately within each group, or cluster classes based on model scores, yielding prediction sets that have better group conditional coverage but larger prediction set size than their baselines.

**Hierarchical and structured CP.** Other works incorporate a known label hierarchy, such as a directed acyclic graph, into the conformal prediction framework. (Hengst et al., 2025) and (Zhang et al., 2025) share the objective of controlling the specificity of prediction sets (e.g., the number of leaf labels) alongside efficiency. (Mortier et al., 2025) introduce the notion of representation complexity, defined as the minimum number of nodes whose descendants cover the prediction set, and study its trade-off with efficiency.

**Hierarchical selective classification.** When a hierarchical structure of labels is available, with classes located at the leaves, Goren et al. (2024) extend the conformal prediction framework to achieve hierarchical selective coverage. Their approach identifies a predicted node by starting from the predicted class (which is a leaf) and ascending the hierarchy

until a conformal threshold is met, in a manner similar to the APS procedure (Romano et al., 2020). This method focuses on controlling the trade-off between predictive accuracy and the specificity of the hierarchical prediction.

**Summary.** To our knowledge, no prior work directly addresses the objective of improving semantic coherence within prediction sets without compromising CP efficiency, let alone leveraging class structure to *improve* efficiency. Existing works in clustered, group-conditional, and hierarchical CP focus on coverage within known groups or on relatively uncommon notions of structure. Importantly, these approaches typically yield prediction sets that are larger than the baselines. In contrast, our approach reduces the number of semantically distinct groups, decreases the average set size, and is applicable across datasets and CP score functions. Moreover, our model-specific approach does not require any prior knowledge of class structure.

## 3. Preliminaries on Conformal Prediction

Let us present notations that are used in the paper, followed by some preliminaries on CP. We consider a $C$-classes classification task of the data $(X, Y)$ distributed on $\mathcal{X} \times [C]$, where $[C] := \{1, \ldots, C\}$. The task is addressed by a classifier model (e.g., a trained deep neural network) that for each input sample $x \in \mathcal{X}$ produces a post-softmax vector $\hat{\pi}(x) \in \mathbb{R}^C$. The predicted class is given by $\hat{y}(x) = \arg\max_i \hat{\pi}_i(x)$.

Conformal Prediction (CP) is a methodology for reliable classification, independent of the data distribution. Given a black-box classifier, predefined $\alpha \in (0, 1)$, and a sample $X$, it generates a *prediction set* of classes, $\mathcal{C}(X)$, such that $Y \in \mathcal{C}(X)$ with probability $1 - \alpha$, where $Y$ is the true class associated with $X$ (Vovk et al., 1999; 2005; Papadopoulos et al., 2002). The decision rule is based on a calibration set of labeled samples $\{x_i, y_i\}_{i=1}^n$. The only assumption in CP is that the random variables associated with the calibration set and the test samples are exchangeable (e.g., the samples are i.i.d.).

Let us state the general process of conformal prediction given the calibration set $\{x_i, y_i\}_{i=1}^n$ and its deployment for a new (test) sample $x_{n+1}$ (for which $y_{n+1}$ is unknown), as presented in (Angelopoulos and Bates, 2021):

1. Define a heuristic score function $s(x, y) \in \mathbb{R}$ based on some output of the model. A higher score should encode a lower level of agreement between $x$ and $y$.

2. Compute $\hat{q}$ as the $\lceil (n + 1)(1 - \alpha) \rceil / n$ quantile of the scores $\{s(x_1, y_1), \ldots, s(x_n, y_n)\}$.

3. At deployment, create the prediction set of a test sample as $\mathcal{C}(x_{n+1}) = \{y : s(x_{n+1}, y) \leq \hat{q}\}$.

CP methods possess the following coverage guarantee.

**Theorem 3.1** (Theorem 1 in (Angelopoulos and Bates, 2021))**.** *Suppose that $\{(X_i, Y_i)\}_{i=1}^n$ and $(X_{n+1}, Y_{n+1})$ are i.i.d., and define $\hat{q}$ as in step 2 above and $\mathcal{C}(X_{n+1})$ as in step 3 above. Then, $\mathbb{P}(Y_{n+1} \in \mathcal{C}(X_{n+1})) \geq 1 - \alpha$.*

The proof of this result is based on (Vovk et al., 1999). A proof of an upper bound of $1 - \alpha + 1/(n + 1)$ also exists (assuming all scores being different almost surely). Note that the coverage is marginal: the probability is taken over the entire distribution of $(X, Y)$ and there is no guarantee per value of $X_{n+1}$.

Different CP methods typically differ by their choice of score function $s(x, y)$, and a key property that they are judged according to is their average prediction set size, $\mathbb{E}[|\mathcal{C}(X)|]$, often refers to as *efficiency*.

## 4. Enhancing CP Using Class Similarity

In this section, we explore the properties of CP when utilizing a known partition of the $C$ classes into $G$ groups. Let $g : [C] \to [G]$ denote the map of classes to groups. Namely, $g(y) \in [G]$ is the index of the group that contains class $y \in [C]$.

As discussed in Section 1, we assume that the groups are superclasses with some semantic meaning. For example, the classes may be cities and the superclasses are geographical location, or the classes are types of diseases and the superclasses group those that require a similar treatment. In this case, it is reasonable for a user to prefer $\mathcal{C}(X)$ whose classes belong only to a few groups, or ideally just to the group of the true label $Y$.

Motivated by the above, let us set a "distance function" between classes based on their groups. Specifically, we consider the binary penalty function given by:

$$d(y, y') := \mathbb{I}\{g(y) \neq g(y')\}, \qquad (1)$$

where $\mathbb{I}\{\cdot\}$ is the indicator function. That is, $d(y, y') = 0$ if $y$ and $y'$ belong to the same group, and otherwise $d(y, y') = 1$. For brevity, we omit the explicit dependence of $d(y, y')$ on $g$.

Given a sample $x$, all common CP methods preserve the ranking of the softmax vector $\hat{\pi}(x)$ and, in particular, include the estimated class $\hat{y}(x)$ in the prediction set before including any other class. Therefore, to reduce the number of groups in $\mathcal{C}(X)$ we propose to penalize a given score function $s(x, y)$ by $d(y, \hat{y}(x))$:

$$s_\lambda(x, y) := s(x, y) + \lambda d(y, \hat{y}(x)), \qquad (2)$$

where $\lambda > 0$ is a parameter. In words, the score of a candidate $y$ that is "semantically far" from $\hat{y}(x)$ is penalized by a value of $\lambda$.

We turn to theoretically explore the properties of CP with $s_\lambda(x, y)$. Let us denote by $\hat{q}_\lambda$ and $\mathcal{C}_\lambda(x)$ the CP threshold and prediction set when using $\lambda > 0$. All the proofs are provided in Appendix A.

**The coverage property is maintained.** This follows directly from Theorem 3.1, as $s_\lambda$ is a valid score that preserves the exchangeability of the calibration and test samples.

**The number of out-of-group labels in the prediction set cannot increase.** To show that adding the penalty term to the score cannot increase the number of classes in $\mathcal{C}_\lambda(x)$ whose group is not $g(\hat{y}(x))$, we first establish the following lemma on the relation between $\hat{q}_\lambda$ and $\hat{q}$.

**Lemma 4.1.** *We have $\hat{q} \leq \hat{q}_\lambda \leq \hat{q} + \lambda$.*

We now present our result on the inclusion of out-of-group labels in the prediction set.

**Proposition 4.2.** *Let $\mathcal{Y}_1(x) := \{y : d(y, \hat{y}(x)) \neq 0\}$. For any $x$ and $\lambda > 0$ we have*

$$\mathcal{C}_\lambda(x) \cap \mathcal{Y}_1(x) \subseteq \mathcal{C}(x) \cap \mathcal{Y}_1(x).$$

The proposition shows that the penalization cannot add any "far"/group-mismatched labels (w.r.t. $\hat{y}(x)$) that weren't already in the unpenalized CP; it can only remove them. This property naturally translates to decreasing the distance-weighted size. Specifically, define $S_\lambda(x) := \sum_{y=1}^{C} d(y, \hat{y}(x)) \mathbb{I}\{y \in \mathcal{C}_\lambda(x)\}$. We have $S_\lambda(x) \leq S_0(x)$ for any $x$, and thus also $\mathbb{E}[S_\lambda(X)] \leq \mathbb{E}[S_0(X)]$. Similarly, eliminating the pathological case of empty $\mathcal{C}(x)$, the number of groups cannot increase, as shown in the following corollary.

**Corollary 4.3.** *Let $\mathcal{G}_\lambda(x)$ and $\mathcal{G}(x)$ denote the groups represented in $\mathcal{C}_\lambda(x)$ and $\mathcal{C}(x)$, respectively. For any $x$ such that $\hat{y}(x) \in \mathcal{C}(x)$ and $\lambda > 0$ we have $\mathcal{G}_\lambda(x) \subseteq \mathcal{G}(x)$.*

Since the corollary holds for any $x$, it reflects the relation $\mathbb{E}[|\mathcal{G}_\lambda(X)|] \leq \mathbb{E}[|\mathcal{G}(X)|]$. This theory supports the empirical observation (in Section 6) that the empirical expectations obey $\hat{\mathbb{E}}[|\mathcal{G}_\lambda(X)|] < \hat{\mathbb{E}}[|\mathcal{G}(X)|]$, with a substantial margin.

**Surprising behavior: The average prediction set size also decreases in practice.** As will be shown in our experiments, with well-tuned $\lambda$ (small enough), we observe $\hat{\mathbb{E}}[|\mathcal{C}_\lambda(X)|] < \hat{\mathbb{E}}[|\mathcal{C}(X)|]$, in benchmark settings, even though the penalty does not imply it directly. Actually, while we reached a guarantee for decreasing the number of "out-of-group" labels, potentially, there can be an increase in "in-group" labels, since $\hat{q} \leq \hat{q}_\lambda$ but the score of $y$ from the same group of $\hat{y}(x)$ remains the same. This case is illustrated in Figure 1.

We turn to establish a theory for reduction in the average prediction set size for small enough $\lambda$. To this end, let us start by some definitions.

**Definition 4.4.** *Given a sample $x$, we have the following definitions:*

1. *"In-group" classes: $\mathcal{Y}_0(x) := \{y : d(y, \hat{y}(x)) = 0\}$ and $n_0(x) := |\mathcal{Y}_0(x)|$.*

2. *"Out-of-group" classes: $\mathcal{Y}_1(x) := \{y : d(y, \hat{y}(x)) \neq 0\}$ and $n_1(x) := |\mathcal{Y}_1(x)|$.*

3. *Per-$x$ conditional quasi-CDF:[1] for $z \in \{0, 1\}$, $\hat{F}_z^x(t) :=$*
$$\frac{1}{n_z(x)} \sum_{y \in \mathcal{Y}_z(x)} \mathbb{I}\{s(x, y) \leq t\}.$$

*We also make the following definitions related to the marginal distribution:*

4. *Average number of "in-group" classes: $\overline{n}_0 := \mathbb{E}[n_0(X)]$.*

5. *Average number of "out-of-group" classes: $\overline{n}_1 := \mathbb{E}[n_1(X)]$.*

6. *Probability of "in-group" true label: $p_0 = \mathbb{P}(Y \in \mathcal{Y}_0(X))$.*

7. *Probability of "out-of-group" true label: $p_1 = \mathbb{P}(Y \in \mathcal{Y}_1(X)) = 1 - p_0$.*

8. *Conditional CDFs: for $z \in \{0, 1\}$, $F_z(t) := \mathbb{P}(s(X, Y) \leq t | Y \in \mathcal{Y}_z(X))$.*

We now state assumptions that will be used in our theorem.

**Assumption 1.** *For small $\lambda \geq 0$, the prediction set $\mathcal{C}_\lambda(X)$ is based on the statistical quantile $q_\lambda$ of the CDF of $s_\lambda(X, Y)$. That is, $q_\lambda$ obeys $\mathbb{P}(s_\lambda(X, Y) \leq q_\lambda) = 1 - \alpha$.*

**Assumption 2.** *For $z \in \{0, 1\}$, the CDF $F_z(t)$ is absolutely continuous, so $f_z(t) = F_z'(t)$ is well-defined.*

**Assumption 3.** *For $z \in \{0, 1\}$, the "size-biased" quasi-CDF $\tilde{F}_z(t) := \frac{1}{\overline{n}_z} \mathbb{E}[n_z(X) \hat{F}_z^X(t)]$ is absolutely continuous, so $\tilde{f}_z(t) = \tilde{F}_z'(t)$ is well-defined.*

Assumptions 1-3 are required for making the analysis tractable, ensuring that $\mathbb{E}[|\mathcal{C}_\lambda(X)|]$ is differentiable with respect to $\lambda$, and sparing cumbersome analysis of the effect of finite calibration sets on inclusion of a label in the predictions sets. Note that Assumption 1 essentially reflects having a large calibration set. Now we present a theorem that characterizes the effect of the penalty with small $\lambda$ on the efficiency.

**Theorem 4.5.** *Consider Definition 4.4. Under Assumptions 1-3, we have*

$$\text{sign}\left(\frac{\mathrm{d}}{\mathrm{d}\lambda} \mathbb{E}[|\mathcal{C}_\lambda(X)|]\Big|_{\lambda=0}\right) = \text{sign}\left(a p_1 \overline{n}_0 - b p_0 \overline{n}_1\right),$$
(3)

*where $a := \tilde{f}_0(q_0) f_1(q_0)$ and $b := \tilde{f}_1(q_0) f_0(q_0)$.*

---

[1] We name the object $\hat{F}_z^x(t)$ "quasi-CDF" because it is not based on any random variable (such as $Y|X = x$) or its realization, but rather on the deterministic set $\mathcal{Y}_z(x)$.

**Discussion.** Let us start by assuming that $a \approx b$. In this case, the theorem shows that the sign of $\frac{\mathrm{d}}{\mathrm{d}\lambda}\mathbb{E}[|\mathcal{C}_\lambda(X)|]\big|_{\lambda=0}$ (where a negative value means that $\lambda \approx 0_+$ reduces $\mathbb{E}[|\mathcal{C}_\lambda(X)|]$) equals the sign of the difference between:

- $p_1 \times \overline{n}_0$: The probability of having the true label out of the group of the predicted class $\times$ the average number of classes in the group of the predicted class.
- $p_0 \times \overline{n}_1$: The probability of having the true label in the group of the predicted class $\times$ the average number of classes out of the group of the predicted class.

We can expect that $p_1\overline{n}_0 \ll p_0\overline{n}_1$ in most practical cases. First, the number of classes within a group is typically much smaller than the number outside the group, implying $\overline{n}_0 \ll \overline{n}_1$, even when groups are not of equal size. For example, in the CIFAR-100 benchmark partition with 20 groups of equal size, we have $\overline{n}_1/\overline{n}_0 = 19$. More generally, unless a single group contains more than half of the classes, this ratio is already guaranteed to exceed 1. Second, for reasonably accurate classifiers, the ratio $p_0/p_1$ is also expected to exceed 1, since $p_0$ corresponds to the group-level accuracy. In particular, top-1 accuracy above $0.5$ already implies $p_0/p_1 > 1$. Therefore, in practical cases if $a \approx b$ or even if $b$ is not much smaller than $a$, then $\mathrm{sign}\,(ap_1\overline{n}_0 - bp_0\overline{n}_1) < 0$. By Theorem 4.5, this implies that penalizing the score with small $\lambda$ will reduce $\mathbb{E}[|\mathcal{C}_\lambda(X)|]$.

Let us discuss the relation between $a$ and $b$. For simplification, assume that the $C$ classes are partitioned to $G$ groups of equal size $K$. In this case, we have constants $n_0(X) = K$ and $n_1(X) = (G-1)K$. So, $\overline{n}_0 = K$ and $\overline{n}_1 = (G-1)K$, as well. Recalling the definition of $\tilde{F}_z(t)$ in Assumption 3, we have

$$\tilde{F}_z(t) = \mathbb{E}[\hat{F}_z^X(t)] = \frac{1}{\overline{n}_z}\mathbb{E}\left[\sum_{y\in\mathcal{Y}_z(X)}\mathbb{I}\{s(X,y)\leq t\}\right].$$

Define $p_z(x) := \mathbb{P}(Y \in \mathcal{Y}_z(x)|X = x)$ and $F_z^x(t) := \mathbb{P}(s(x,Y) \leq t|Y \in \mathcal{Y}_z(x), X = x)$. Observe that

$$F_z^x(t) = \frac{\mathbb{P}(s(x,Y)\leq t, Y\in\mathcal{Y}_z(x)|X=x)}{\mathbb{P}(Y\in\mathcal{Y}_z(x)|X=x)}$$
$$= \frac{\sum_{y\in\mathcal{Y}_z(x)}\mathbb{P}(Y=y|X=x)\mathbb{I}\{s(x,y)\leq t\}}{p_z(x)}.$$

Using the relation (see derivation in Appendix A.5):

$$F_z(t) = \frac{1}{p_z}\mathbb{E}[p_z(X)F_z^X(t)] \tag{4}$$

and substituting the expression derived for $F_z^x(t)$, we get

$$F_z(t) = \frac{1}{p_z}\mathbb{E}[p_z(X)F_z^X(t)]$$
$$= \frac{1}{p_z}\mathbb{E}\left[\sum_{y\in\mathcal{Y}_z(X)}\mathbb{P}(Y=y|X)\mathbb{I}\{s(X,y)\leq t\}\right].$$

Thus, if, per $X = x$, the labels $Y \in \mathcal{Y}_z(x)$ are distributed uniformly, then the factor that multiplies $\mathbb{I}\{s(X,y) \leq t\}$ is constant, and therefore $\tilde{F}_z(t) = F_z(t)$. This gives exact $a = b$ (recall their definitions in Theorem 4.5), so the above arguments hold for having $\mathrm{sign}\,(ap_1\overline{n}_0 - bp_0\overline{n}_1) < 0$. The fact that the theory includes integration over $X$ and considers the densities only at $q_0$, together with the empirical fact that $p_1\overline{n}_0 \ll p_0\overline{n}_1$, teach us that there are many cases where $\mathrm{sign}\,(ap_1\overline{n}_0 - bp_0\overline{n}_1) < 0$ even for non uniform conditional label distributions.

Notably, Theorem 4.5 also clarifies when the penalty may fail to improve efficiency: in extreme cases such as highly dominant groups and very weak models, where $\mathrm{sign}\,(ap_1\overline{n}_0 - bp_0\overline{n}_1) > 0$. Yet, such extreme cases have not been encountered in any of the many benchmark setups that we examined. A limitation of the theorem is that it is a local result for $\lambda$ near 0. As empirically shown in Section 6, the range of values of $\lambda$ for which the average prediction set size decreases is not small, which highlights the robustness of our penalty method.

## 5. Extension to Model-Specific Class Similarity

The original motivation for the penalized score in equation 2 came from considering the potential preference of the user to reduce the number of "semantically far" classes in the prediction set. However, Theorem 4.5 reveals that, perhaps surprisingly, the proposed penalty has a beneficial effect on the prediction set size for any score function, provided that the penalty parameter $\lambda$ is sufficiently small and $p_1\overline{n}_0 \ll p_0\overline{n}_1$ (omitting the effect of $a$ and $b$ in equation 3).

This result actually tells us that, in terms of efficiency, we can gain more from partitions into groups that are as small as possible (low $\overline{n}_0$ and high $\overline{n}_1$), as long as the probability of making out-of-group mistakes ($p_1 = 1 - p_0$) is kept low. Nothing in this result requires a human-related semantic similarity between classes within a group. This motivates us to propose a *model-specific* extension of the method. Specifically, given a pretrained classifier, we suggest basing the penalty on the class similarity *perceived by the model*. An important advantage of this extension, which focuses on boosting efficiency rather than group-related metrics, is that it eliminates the need for a human-made semantic partition, which may not be available for some datasets.

For a given classifier, the proposed extension requires computing a $C \times C$ class similarity matrix, denoted by $M$ (recall that $C$ denotes the number of classes). The $(c, c')$ entry in $M$ should reflect the similarity between classes $c$ and $c'$, as perceived by the model. Similarity metrics are typically continuous, e.g., inner products and kernels. Binarization of such metrics will require tuning a threshold parameter. Hence, we propose to diverge slightly from the

method in Section 4 by allowing the similarity metric to be "soft", which also adds more flexibility to the method. For $M_{c,c'} \in \mathbb{R}$, upper bounded by 1 as the maximal similarity, we define the soft model-specific penalty function:

$$d^{MS}(y, y') := 1 - M_{y,y'}. \tag{5}$$

Substituting this penalty in equation 2 in lieu of the model-agnostic $d$, gives

$$s_\lambda^{MS}(x, y) := s(x, y) + \lambda d^{MS}(y, \hat{y}(x)). \tag{6}$$

**Determining model-specific class similarity.** There exist multiple potential strategies for constructing a class similarity matrix $M$ given a model. Here, we propose one that consistently improves the efficiency results in our experiments. Future research may attempt to optimize this choice. We assume access to the labeled training samples. The last layer of a deep neural network-based classifier $f(x) \in \mathbb{R}^C$, before the softmax operation, can be typically expressed as: $f(x) = W h_\theta(x) + b$, where $h_\theta(\cdot) : \mathcal{X} \to \mathbb{R}^p$ (with $p \geq C$) is the deepest feature mapping that is composed of all the hidden layers (with learnable parameters $\theta$), and $W \in \mathbb{R}^{C \times p}$ and $b \in \mathbb{R}^C$ are the weights and bias of the last classification layer. We determine the class similarity according to a similarity function between the means of different classes in the deepest feature space. This strategy is motivated by recent work on the neural collapse phenomenon (Papyan et al., 2020), where the within-class samples of well-trained classifiers concentrate around their class mean in feature space, while inter-class means are well separated yet empirically still preserve relations that generalize to test data (Tirer et al., 2023; Yang et al., 2023). Therefore, examining the relation between class means in feature space yields small effective groups without compromising on group-wise accuracy.

Denote by $\{x_{c,i}\}$, $i \in [n_c]$, the training samples associated with class $c \in [C]$. Compute the class means and the global mean of the features:

$$\overline{h}_c = \frac{1}{n_c} \sum_{i=1}^{n_c} h_\theta(x_{c,i}), \qquad \overline{h}_G = \frac{1}{C} \sum_{c=1}^{C} \overline{h}_c.$$

We then set the entry $M_{c,c'}$ in the class similarity matrix $M$ using the cosine similarity of the centered class means:

$$M_{c,c'} = \frac{\langle \overline{h}_c - \overline{h}_G, \overline{h}_{c'} - \overline{h}_G \rangle}{\|\overline{h}_c - \overline{h}_G\| \|\overline{h}_{c'} - \overline{h}_G\|}.$$

**Remark.** Note that the proposed penalized score $s_\lambda^{MS}$ preserves the exchangeability of the calibration and test samples, and hence the coverage property is maintained (Theorem 3.1). The model-specific penalty is directly motivated by Theorem 4.5, but shifts from a binary penalty to a continuous one. We find that extending Theorem 4.5 to this more general setting is technically challenging. We believe

that this is an interesting direction for future research, as establishing such a result may also help formally explain the benefits of certain score-specific regularization methods, such as the improved efficiency of RAPS compared to APS.

**Guidance for choosing between the model-agnostic and model-specific variants.** As will be shown in Section 6, both our model-agnostic variant (presented in Section 4) and our model-specific variant (presented in this section) outperform the baselines in terms of efficiency. The latter further reduces the prediction set size by adapting to the model's representations, which strengthens the main message of this paper: leveraging class similarity can significantly improve conformal prediction efficiency, especially when the similarity reflects the model's perception of the data. Another advantage of the model-specific variant is eliminating the need for a human semantic partition, which makes it applicable to any dataset. Yet, there are also natural reasons to choose the model-agnostic variant when a reliable group structure is given: it is accompanied by more established theoretical support (in particular for reduction in the number of groups), and it is applicable to black-box models and does not require access to training data.

## 6. Experiments

**Datasets and models.** We conduct experiments on four image classification benchmarks: CIFAR-100 (Krizhevsky et al., 2009), Living-17 from the BREEDS suite (Santurkar et al., 2020), ImageNet with 1K classes (Deng et al., 2009); and Mini-ImageNet (Vinyals et al., 2016), a subset of ImageNet with 100 classes. Note that CIFAR-100 and Living-17 have official semantic superclass structures, i.e., partitions of the classes into coarse classes. Specifically, CIFAR-100 has 20 superclasses (e.g., aquatic mammals, fish, flowers, etc.), where each one groups 5 classes (e.g., beaver, dolphin, otter, seal, and whale are grouped under aquatic mammals). Similarly, Living-17 has 17 superclasses, where each one groups 4 classes. We use ResNet50 (He et al., 2016) as the classifier model for all datasets except ImageNet, where we use ViT-B/16 (Dosovitskiy et al., 2021). For CIFAR-100 we use ResNet34 as well. Details on the training of the models are provided in Appendix B.1. We split the validation sets of the datasets to 20% calibration and 80% test, as common in the literature.

**CP score functions.** We consider three prominent conformal score functions $s(x, y)$: (1) LAC (Sadinle et al., 2019), defined as one minus the classifier's softmax value at index $y$; (2) RAPS (Angelopoulos et al., 2021), based on cumulating softmax entries up to the rank of $y$, like APS (Romano et al., 2020), but includes a regularization term that yields smaller prediction sets; and (3) SAPS (Huang et al., 2024), penalizes the maximal softmax entry according to the rank of $y$. Detailed definitions of these scores are provided in Ap-

*Table 1.* Performance comparison of various CP methods with $\alpha = 0.05$.

| | #Superclasses ↓ | | | Size ↓ | | | | |
|---|---|---|---|---|---|---|---|---|
| Method | CIFAR100, RN50 | CIFAR100, RN34 | L17, RN50 | ImageNet, ViT | m-ImageNet, RN50 | CIFAR100, RN50 | CIFAR100, RN34 | L17, RN50 |
| **LAC** | | | | | | | | |
| Standard | 2.27 ($\pm$0.276) | 2.41 ($\pm$0.274) | 1.26 ($\pm$0.068) | 1.88 ($\pm$0.216) | 4.73 ($\pm$0.767) | 3.68 ($\pm$0.759) | 3.82 ($\pm$0.707) | 1.77 ($\pm$0.205) |
| Clustered | 2.28 ($\pm$0.248) | 2.34 ($\pm$0.200) | 1.24 ($\pm$0.035) | 2.73 ($\pm$0.978) | 4.50 ($\pm$0.823) | 3.70 ($\pm$0.676) | 3.62 ($\pm$0.485) | 1.69 ($\pm$0.101) |
| AIR | **1.36** ($\pm$0.097) | **1.43** ($\pm$0.091) | **1.16** ($\pm$0.040) | N/A | N/A | 6.80 ($\pm$0.101) | 7.15 ($\pm$0.089) | 5.80 ($\pm$0.061) |
| MA-CS | 1.85 ($\pm$0.183) | 1.92 ($\pm$0.108) | 1.19 ($\pm$0.068) | N/A | N/A | 3.17 ($\pm$0.424) | 3.51 ($\pm$0.749) | 1.71 ($\pm$0.183) |
| MS-CS | 1.83 ($\pm$0.137) | 1.87 ($\pm$0.126) | 1.19 ($\pm$0.058) | **1.80** ($\pm$0.161) | **3.82** ($\pm$0.696) | **2.92** ($\pm$0.339) | **2.94** ($\pm$0.339) | **1.70** ($\pm$0.156) |
| **RAPS** | | | | | | | | |
| Standard | 2.49 ($\pm$0.145) | 3.40 ($\pm$0.112) | 1.35 ($\pm$0.052) | 3.27 ($\pm$0.176) | 8.67 ($\pm$2.398) | 3.83 ($\pm$0.276) | 5.79 ($\pm$0.223) | 1.98 ($\pm$0.167) |
| Clustered | 2.42 ($\pm$0.103) | 3.32 ($\pm$0.115) | 1.33 ($\pm$0.027) | 4.25 ($\pm$2.505) | 8.19 ($\pm$2.224) | 3.67 ($\pm$0.200) | 5.62 ($\pm$0.232) | 1.90 ($\pm$0.090) |
| AIR | **1.95** ($\pm$0.077) | 3.03 ($\pm$0.085) | **1.20** ($\pm$0.026) | N/A | N/A | 9.75 ($\pm$0.121) | 15.15 ($\pm$0.095) | 6.00 ($\pm$0.051) |
| MA-CS | 2.01 ($\pm$0.160) | 2.29 ($\pm$0.250) | 1.23 ($\pm$0.081) | N/A | N/A | 3.50 ($\pm$0.305) | 4.52 ($\pm$0.501) | 1.97 ($\pm$0.295) |
| MS-CS | **1.95** ($\pm$0.122) | **2.22** ($\pm$0.182) | 1.24 ($\pm$0.050) | **2.55** ($\pm$0.284) | **7.35** ($\pm$2.495) | **3.17** ($\pm$0.265) | **3.79** ($\pm$0.387) | **1.81** ($\pm$0.222) |
| **SAPS** | | | | | | | | |
| Standard | 2.33 ($\pm$0.242) | 2.39 ($\pm$0.17) | 1.32 ($\pm$0.048) | 2.13 ($\pm$0.107) | 5.93 ($\pm$1.480) | 3.45 ($\pm$0.458) | 3.57 ($\pm$0.342) | 1.94 ($\pm$0.164) |
| Clustered | 2.55 ($\pm$0.293) | 2.62 ($\pm$0.276) | 1.31 ($\pm$0.045) | 10.0 ($\pm$4.351) | 8.03 ($\pm$2.173) | 3.86 ($\pm$0.547) | 4.02 ($\pm$0.521) | 2.01 ($\pm$0.172) |
| AIR | **1.51** ($\pm$0.163) | **1.59** ($\pm$0.124) | **1.11** ($\pm$0.032) | N/A | N/A | 7.55 ($\pm$0.324) | 7.95 ($\pm$0.274) | 5.55 ($\pm$0.062) |
| MA-CS | 1.88 ($\pm$0.286) | 1.26 ($\pm$0.033) | 1.93 ($\pm$0.162) | N/A | N/A | **3.14** ($\pm$0.464) | **3.32** ($\pm$0.271) | 1.94 ($\pm$0.147) |
| MS-CS | 1.97 ($\pm$0.187) | 2.14 ($\pm$0.175) | 1.24 ($\pm$0.035) | **2.08** ($\pm$0.110) | **4.70** ($\pm$1.026) | **3.14** ($\pm$0.361) | **3.32** ($\pm$0.371) | **1.87** ($\pm$0.146) |

pendix B.2. We conduct experiments using target coverage levels of $\alpha = \{0.05, 0.1\}$, as common in the literature.

**Details of the CP methods evaluated.** For each score function, we evaluate the following versions.

- Standard: The CP algorithm with the original score function and no modifications.
- Clustered (Ding et al., 2023): The algorithm extends and improves the efficiency of class-wise Mondrian CP (Vovk, 2012) (which applies CP separately to each class) by grouping classes into $m$ clusters based on the similarity of score distributions, and applying CP on each cluster. The algorithm has two parameters: $\gamma$, which controls the proportion of data used for clustering, and $m$, the number of clusters. We set $\gamma = 0.2$ to match the proportion used in our methods. $m$ is chosen following (Ding et al., 2023). Details can be found in Appendix B.4 of their paper.
- AIR (Accumulating Inference Rule): Inspired by the *Climbing Inference Rule* (Goren et al., 2024), which climbs the hierarchy from a predicted leaf node to its parent until reaching a conformalized threshold that guarantees coverage. The exact approach of (Goren et al., 2024) does not suit the two-level superclass structure of CIFAR-100 and Living-17, as it often leads to the inclusion of all classes. To address this, *we develop an improved variant*. Instead of climbing to the parent, it accumulates mass onto the next superclass with the highest probability, effectively applying conformal prediction at the superclass level rather than the class level.
- MA-CS (Model-Agnostic Class-Similarity): The standard CP algorithm augmented with our binary regularization term, as described in Section 4. To select the regularization parameter $\lambda$, we split the calibration set into two equal size sets: $\hat{q}$-calibration (used to compute $\hat{q}$), and $\lambda$-evaluation (used to evaluate performance for different $\lambda$ values). We iterate over a predefined set of $\lambda$ values and choose the one that achieves the smallest set size on the $\lambda$-evaluation set.

- MS-CS (Model-Specific Class-Similarity): The standard CP algorithm combined with our regularization term based on the model-specific similarity matrix, as detailed in Section 5. The regularization parameter $\lambda$ is set using the same procedure as in MA-CS.

Note that AIR and our MA-CS cannot be applied for ImageNet and Mini-ImageNet, which lacks a pre-specified superclass structure. Yet, our MS-CS remains applicable.

**Evaluation metrics.** The evaluation metrics that we use are the average prediction set size, and for CIFAR-100 and Living-17 also the average number of superclasses in the prediction set. Note that for these metrics: *the lower the better*. The metrics are computed over the test set and we report their means and standard deviations based on 100 trials (random splits of 20% calibration set and 80% test set). We also compute the marginal coverage and a metric for the worst class-conditional coverage gap. The definitions of the metrics are stated in Appendix B.3.

The code for reproducing our experiments is available at https://github.com/ariel361/CP_via_CS.

### 6.1. Results

We begin with reporting the top-1 accuracy of each of the four dataset-model pairs: ViT on ImageNet: 83.91%; ResNet50 on Mini-ImageNet: 80.38%; ResNet50 on CIFAR-100: 80.93%; ResNet34 on CIFAR-100: 78.92%; ResNet50 on Living-17: 84.68%.

Due to space limitation, the coverage metrics results are reported in the Appendix. In Appendix C.1 (Tables 3 and 4), we report the marginal coverage of our MA-CS and MS-CS methods. The pre-specified coverage level of $1 - \alpha$ is preserved. In this appendix we report the marginal coverage of the other methods, which also satisfy the specified level.

In Appendix C.2 (Tables 5 and 6), we report the worst class-conditional coverage gap for each method across all settings.

*Table 2.* Performance comparison of various CP methods with $\alpha = 0.1$.

| Method | #Superclasses ↓ | | | Size ↓ | | | | |
|---|---|---|---|---|---|---|---|---|
| | CIFAR100, RN50 | CIFAR100, RN34 | L17, RN50 | ImageNet ViT | m-ImageNet, RN50 | CIFAR100, RN50 | CIFAR100, RN34 | L17, RN50 |
| **LAC** | | | | | | | | |
| Standard | 1.37 ($\pm$0.046) | 1.44 ($\pm$0.051) | 1.07 ($\pm$0.023) | 1.24 ($\pm$0.065) | 1.79 ($\pm$0.130) | 1.62 ($\pm$0.084) | 1.75 ($\pm$0.091) | 1.21 ($\pm$0.064) |
| Clustered | 1.48 ($\pm$0.144) | 1.53 ($\pm$0.128) | 1.09 ($\pm$0.036) | 1.40 ($\pm$0.070) | 2.18 ($\pm$0.379) | 1.82 ($\pm$0.267) | 1.90 ($\pm$0.221) | 1.27 ($\pm$0.083) |
| AIR | **1.02** ($\pm$0.097) | **1.09** ($\pm$0.091) | 1.06 ($\pm$0.040) | N/A | N/A | 5.10 ($\pm$0.101) | 5.45 ($\pm$0.089) | 5.30 ($\pm$0.061) |
| MA-CS | 1.24 ($\pm$0.063) | 1.34 ($\pm$0.065) | **1.04** ($\pm$0.018) | N/A | N/A | 1.54 ($\pm$0.127) | 1.70 ($\pm$0.120) | 1.19 ($\pm$0.037) |
| MS-CS | 1.25 ($\pm$0.053) | 1.32 ($\pm$0.072) | 1.05 ($\pm$0.015) | **1.21** ($\pm$0.056) | **1.69** ($\pm$0.158) | **1.53** ($\pm$0.092) | **1.65** ($\pm$0.138) | **1.18** ($\pm$0.042) |
| **RAPS** | | | | | | | | |
| Standard | 1.92 ($\pm$0.053) | 2.69 ($\pm$0.062) | 1.19 ($\pm$0.019) | 2.48 ($\pm$0.100) | 3.79 ($\pm$0.142) | 2.70 ($\pm$0.102) | 4.33 ($\pm$0.125) | 1.51 ($\pm$0.049) |
| Clustered | 1.91 ($\pm$0.087) | 2.64 ($\pm$0.108) | 1.20 ($\pm$0.098) | 2.46 ($\pm$0.081) | 4.28 ($\pm$0.661) | 2.69 ($\pm$0.166) | 4.22 ($\pm$0.222) | 1.54 ($\pm$0.122) |
| AIR | 1.52 ($\pm$0.077) | 1.63 ($\pm$0.085) | 1.10 ($\pm$0.026) | N/A | N/A | 7.60 ($\pm$0.121) | 8.15 ($\pm$0.095) | 5.50 ($\pm$0.051) |
| MA-CS | **1.34** ($\pm$0.099) | **1.45** ($\pm$0.100) | **1.03** ($\pm$0.032) | N/A | N/A | 2.10 ($\pm$0.177) | 2.60 ($\pm$0.165) | 1.38 ($\pm$0.053) |
| MS-CS | **1.34** ($\pm$0.085) | 1.49 ($\pm$0.080) | 1.07 ($\pm$0.020) | **1.44** ($\pm$0.122) | **2.05** ($\pm$0.203) | **1.89** ($\pm$0.160) | **2.18** ($\pm$0.174) | **1.28** ($\pm$0.056) |
| **SAPS** | | | | | | | | |
| Standard | 1.48 ($\pm$0.110) | 1.57 ($\pm$0.096) | 1.10 ($\pm$0.017) | 1.52 ($\pm$0.020) | 2.16 ($\pm$0.395) | 1.83 ($\pm$0.204) | 1.97 ($\pm$0.186) | 1.29 ($\pm$0.044) |
| Clustered | 1.60 ($\pm$0.265) | 1.73 ($\pm$0.299) | 1.20 ($\pm$0.224) | 2.93 ($\pm$0.081) | 2.96 ($\pm$0.760) | 1.99 ($\pm$0.336) | 2.18 ($\pm$0.406) | 1.49 ($\pm$0.236) |
| AIR | **1.16** ($\pm$0.039) | **1.10** ($\pm$0.023) | **1.02** ($\pm$0.017) | N/A | N/A | 5.80 ($\pm$0.207) | 5.50 ($\pm$0.155) | 4.03 ($\pm$0.145) |
| MA-CS | 1.26 ($\pm$0.044) | 1.42 ($\pm$0.063) | 1.06 ($\pm$0.012) | N/A | N/A | 1.74 ($\pm$0.140) | 1.89 ($\pm$0.163) | 1.27 ($\pm$0.062) |
| MS-CS | 1.36 ($\pm$0.084) | 1.39 ($\pm$0.059) | 1.07 ($\pm$0.013) | **1.37** ($\pm$0.017) | **1.95** ($\pm$0.239) | **1.71** ($\pm$0.172) | **1.81** ($\pm$0.136) | **1.24** ($\pm$0.049) |

Overall, our class-similarity approach does not significantly change this metric compared to the standard versions. In fact, for LAC and SAPS, our MS-CS often yields modest improvement (smaller gaps from $1 - \alpha$).

In Tables 1 and 2 we report the results for the average prediction set size and, when relevant, the average number of superclasses in the prediction set for coverage levels of $\{0.05, 0.1\}$ respectively. Both tables contain similar findings. Let us discuss the results of Table 1.

**Comparison between our methods and Standard/Clustered.** Excluding AIR (discussed separately), our methods—MA-CS and MS-CS—consistently achieve the best performance on both metrics across all dataset–model pairs and all CP methods. For example, on CIFAR100–ResNet34 with RAPS score, MA-CS and MS-CS obtain average set size of 4.52 and 3.79, compared to 5.79 and 5.62 for Standard and Clustered, representing a reduction of more than $30\%$. Similarly, they achieve #Superclasses values of 2.29 and 2.22, whereas Standard and Clustered yield 3.40 and 3.32, corresponding to reductions of approximately $33\%$.

**Comparison between our methods and AIR.** For the #Superclasses metric, performance varies across score functions and AIR often achieves lower values. For example, on CIFAR100–ResNet50 with the SAPS score, MA-CS and MS-CS achieve #Superclasses values of 1.88 and 1.97, compared to 1.51 for AIR. Yet, importantly, for the average size metric, our methods consistently and significantly outperform AIR across all settings. For instance, on CIFAR100-ResNet34 with the RAPS score, MA-CS and MS-CS achieve values of 4.52 and 3.79, compared to 15.15 for AIR, corresponding to a substantial reduction of approximately $75\%$. Similar reductions are observed throughout the remaining results.

**Comparison between MA-CS and MS-CS.** The two methods achieve similar overall performance, with MS-CS yielding smaller prediction set sizes. This improvement can be attributed to the higher flexibility of MS-CS, which leverages model-specific information and "soft" similarity values. Specifically, it utilizes similarities between classes as perceived by the model itself, while the predefined superclasses used by MA-CS (not used in training) are less aligned with the model's representations and may also be coarser. Interestingly, for #Superclasses the results of MS-CS are similar to those of MA-CS, despite the class similarities being derived automatically from the model. This may be explained by the overall smaller sets of our MS-CS variant, which can naturally lead to classes from fewer superclasses being included.

### 6.2. The effect of $\lambda$ on the metrics

In this section, we examine how the penalty coefficient $\lambda$ affects the performance. In Figure 2, we present the metrics of MA-CS with different values of $\lambda$ for CIFAR-100, ResNet50 and RAPS score with $\alpha = 0.1$. In order to emphasize practical usage, we used only 10% of the data for calibration and another 10% for validation.

Examining #Superclasses (red curve in Figure 2(left)), as expected, for a large range of $\lambda$ it decreases as $\lambda$ increases, reflecting the intended effect of the penalty. The slight increase starting from a very large (impractical) value of $\lambda$ can be explained by a decrease in number of samples with empty prediction sets. Namely, even for samples where the minimal score across labels is quite large, due to overly large $\lambda$ the CP threshold becomes large enough to upper bound the minimal score.

As for the average set size (blue curve in Figure 2(left)), observe that it initially decreases for small values of $\lambda$, aligned with Theorem 4.5. Then, after reaching a minimum, the metric increases with $\lambda$. The existence of a stable range of $\lambda$ in which both metrics exhibit significant improvement underscores the practicality of our tuning approach, which adds no more than a few minutes to the calibration stage.

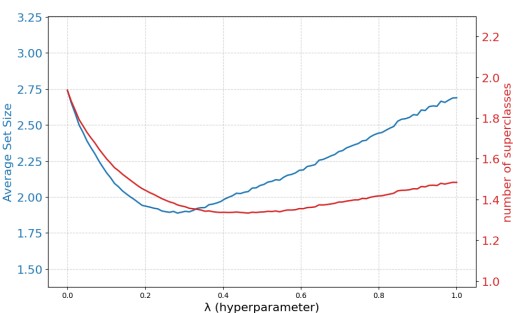 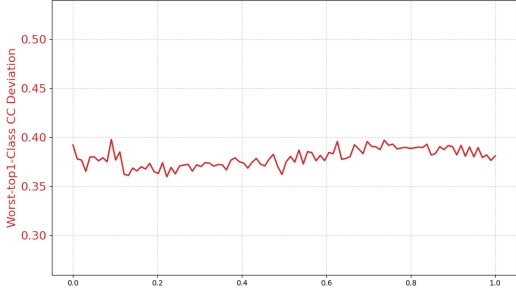

*Figure 2.* The effect of $\lambda$ on the average set size (blue, left), #Superclasses (red, left), and worst class-conditional coverage gap (red, right) for CIFAR-100, ResNet50 and RAPS score.

In Figure 2(right), we present the effect of $\lambda$ on the worst class-conditional coverage gap. As shown, this metric remains relatively stable across the range of $\lambda$ values. Interestingly, varying $\lambda$ does not induce a tradeoff between prediction set size and worst class-conditional coverage, in contrast to other hyperparameters such as temperature scaling (Dabah and Tirer, 2025).

### 6.3. Model-perceived class similarity

In this part, we show the similarity between classes as perceived by the model.

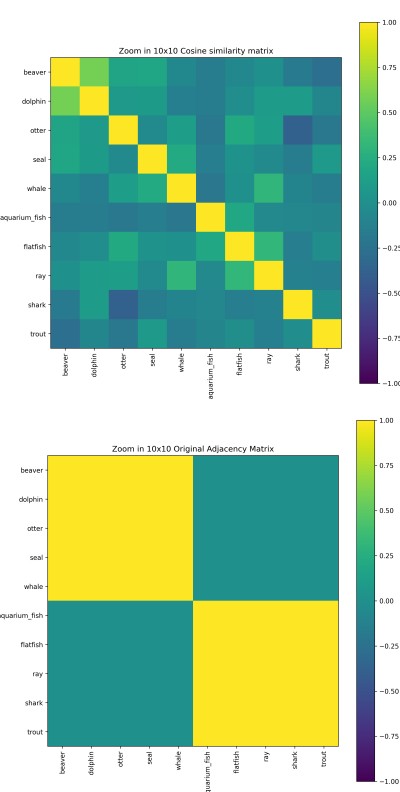

*Figure 3.* Comparison of zoomed 10×10 regions of the model specific (top) and model agnostic (bottom) similarity matrices for CIFAR-100 and ResNet50.

For CIFAR100-ResNet50, we present in Figure 3 the top-left $10 \times 10$ block of the similarity matrix $M$ (defined in section 5). The associated model-agnostic similarity is depicted as well. The complete matrices are provided in Appendix D.

To further highlight the advantage of the model-perceived class similarity, we compare the performance of the MS-CS matrix against the identity matrix $M = I$, which equally punishes all classes except the prediction. The experiments in Appendix D.2 demonstrate the superiority of both MA-CS and MS-CS over this naive variant, suggesting that using the learned embeddings from the model itself is highly valuable. As another ablation, we show in this appendix the performance degradation that results from using random (incorrect) similarity between classes.

## 7. Conclusion

In this paper, we proposed a class-similarity-based regularization approach that can be applied to any CP score function and reduces *both* the number of groups *and* the overall size of the prediction sets. We backed our model-agnostic variant with comprehensive theory, which also motivated us to extend it to a novel model-specific approach. Importantly, the latter reduces the prediction set size even further and does not require any known class structure, making it a widely applicable tool in the CP toolbox.

Future work may extend our theoretical analysis, in particular to cover the model-specific variant, and stress-test the approach on weak classifiers. Our penalty depends on hard or soft group-wise correctness, which can be viewed as a relaxation of top-1 correctness. Although our approach consistently improved CP across all examined benchmarks with standard classifiers, the gains may diminish when the model is too weak, as implied by our theory.

## Acknowledgments

The work is supported by the Israel Science Foundation grant No. 1940/23 and the MOST grant No. 0007091.

# Impact Statement

This paper presents work whose goal is to advance the field of Machine Learning. There are many potential societal consequences of our work, none which we feel must be specifically highlighted here.

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

# A. Proofs and Additional Derivations

### A.1. Proof of Lemma 4.1

For *any* $(x_i, y_i)$ in the calibration set we have $s(x_i, y_i) \leq s_\lambda(x_i, y_i) \leq s(x_i, y_i) + \lambda$. The $(1 - \alpha)$ empirical quantile for $\{s(x_i, y_i)\}$ is $\hat{q}$ and for $\{s(x_i, y_i) + \lambda\}$ is $\hat{q} + \lambda$. Therefore the $(1 - \alpha)$ empirical quantile for $\{s_\lambda(x_i, y_i)\}$ is in $[\hat{q}, \hat{q} + \lambda]$.

### A.2. Proof of Proposition 4.2

For any $y \in \mathcal{Y}_1(x)$, we have $d(y, \hat{y}(x)) = 1$, so the inclusion in $\mathcal{C}_\lambda(x)$ implies satisfying $s(x, y) + \lambda \leq \hat{q}_\lambda \leq \hat{q} + \lambda$, where the second inequality follows from Lemma 4.1. Therefore, $s(x, y) \leq \hat{q}$, which implies $y \in \mathcal{C}(x)$.

### A.3. Proof of Corollary 4.3

Formally, $\mathcal{G}_\lambda(x) = \{g(y) : y \in \mathcal{C}_\lambda(x)\}$ and $\mathcal{G}(x) = \{g(y) : y \in \mathcal{C}(x)\}$. We already have in Proposition 4.2 that any $y$ with $g(y) \neq g(\hat{y}(x))$ cannot be added to $\mathcal{C}_\lambda(x)$. The assumption that $\hat{y}(x) \in \mathcal{C}(x)$ eliminates the pathological case that $s_\lambda(x, \hat{y}(x)) \leq \hat{q}_\lambda$ but $s(x, \hat{y}(x)) > \hat{q}$. So, $g(\hat{y}(x))$ is already in both $\mathcal{G}_\lambda(x)$ and $\mathcal{G}(x)$.

### A.4. Proof of Theorem 4.5

From the definition of $s_\lambda(x, y)$, for any fixed $x$, the size of the penalized conformal set can be written as

$$|\mathcal{C}_\lambda(x)| = \sum_{y \in \mathcal{Y}_0(x)} \mathbb{I}\{s(x, y) \leq q_\lambda\} + \sum_{y \in \mathcal{Y}_1(x)} \mathbb{I}\{s(x, y) \leq q_\lambda - \lambda\}$$
$$= n_0(x)\hat{F}_0^x(q_\lambda) + n_1(x)\hat{F}_1^x(q_\lambda - \lambda),$$

where in the first equation we used $q_\lambda$ due to Assumption 1, and in the second equation we used the definition of $\hat{F}_z^x(t)$. By the law of total expectation,

$$\mathbb{E}[|\mathcal{C}_\lambda(X)|] = \mathbb{E}[n_0(X)\hat{F}_0^X(q_\lambda)] + \mathbb{E}[n_1(X)\hat{F}_1^X(q_\lambda - \lambda)].$$

Using the definition of $\tilde{F}_z(t)$ in Assumption 3, we get

$$\mathbb{E}[|\mathcal{C}_\lambda(X)|] = \overline{n}_0 \, \tilde{F}_0(q_\lambda) + \overline{n}_1 \, \tilde{F}_1(q_\lambda - \lambda). \tag{7}$$

Next, recall that $q_\lambda$ is defined as the $(1 - \alpha)$–quantile of the CDF $F_\lambda(t) := \mathbb{P}(s_\lambda(X, Y) \leq t)$. Namely, $F_\lambda(q_\lambda) = 1 - \alpha$. Observe that

$$F_\lambda(t) = \mathbb{P}(Y \in \mathcal{Y}_0(x))F_0(t) + \mathbb{P}(Y \in \mathcal{Y}_1(x))F_1(t - \lambda)$$
$$= p_0 F_0(t) + p_1 F_1(t - \lambda).$$

Applying implicit differentiation, by differentiating both sides of $F_\lambda(q_\lambda) = 1 - \alpha$ with respect to $\lambda$ (valid due to Assumption 2), we get

$$\frac{\partial F_\lambda}{\partial \lambda}(q_\lambda) + \frac{\partial F_\lambda}{\partial t}(q_\lambda) \frac{dq_\lambda}{d\lambda} = 0.$$

Since

$$\frac{\partial F_\lambda}{\partial t}(t) = p_0 f_0(t) + p_1 f_1(t - \lambda), \quad \frac{\partial F_\lambda}{\partial \lambda}(t) = -p_1 f_1(t - \lambda),$$

we obtain

$$\frac{dq_\lambda}{d\lambda} = \frac{p_1 f_1(q_\lambda - \lambda)}{p_0 f_0(q_\lambda) + p_1 f_1(q_\lambda - \lambda)}.$$

At $\lambda = 0$ (with $q = q_0$),

$$\frac{dq_\lambda}{d\lambda}\bigg|_{\lambda=0} = \frac{p_1 f_1(q)}{p_0 f_0(q) + p_1 f_1(q)}. \tag{8}$$

Now, let us differentiate equation 7 with respect to $\lambda$ (valid due to Assumption 3):

$$\frac{d}{d\lambda} \mathbb{E}[|\mathcal{C}_\lambda(X)|] = \overline{n}_0 \tilde{f}_0(q_\lambda) \frac{dq_\lambda}{d\lambda} + \overline{n}_1 \tilde{f}_1(q_\lambda - \lambda) \left(\frac{dq_\lambda}{d\lambda} - 1\right).$$

Evaluating at $\lambda = 0$ and substituting equation 8,

$$\frac{d}{d\lambda} \mathbb{E}[|\mathcal{C}_\lambda(X)|]\Big|_{\lambda=0} = \overline{n}_0 \tilde{f}_0(q) \frac{p_1 f_1(q)}{p_0 f_0(q) + p_1 f_1(q)} + \overline{n}_1 \tilde{f}_1(q) \left( \frac{p_1 f_1(q)}{p_0 f_0(q) + p_1 f_1(q)} - 1 \right)$$

$$= \frac{1}{p_0 f_0(q) + p_1 f_1(q)} \left( \tilde{f}_0(q) f_1(q) \cdot p_1 \overline{n}_0 - \tilde{f}_1(q) f_0(q) \cdot p_0 \overline{n}_1 \right).$$

Since $\dfrac{1}{p_0 f_0(q) + p_1 f_1(q)} > 0$ (strictly positive), we obtain equation 3.

### A.5. Derivation of equation 4

To simplify the notation, define the event $A_z = Y' \in \mathcal{Y}_z(X)$. We have

$$\begin{aligned}
F_z(t) &= \mathbb{P}(s(X,Y) \leq t | A_z) \\
&= \mathbb{E}_{X,Y|A_z}[\mathbb{I}\{s(X,Y) \leq t\}] \\
&= \mathbb{E}_{X|A_z}[\mathbb{E}_{Y|A_z,X}[\mathbb{I}\{s(X,Y) \leq t\}]] \\
&= \mathbb{E}_{X|A_z}[\mathbb{P}(s(x,Y) \leq t | Y \in \mathcal{Y}_z(X), X)] \\
&= \mathbb{E}_{X|A_z}[F_z^X(t)]
\end{aligned}$$

Next, using $p_{X|A_z}(x) = \dfrac{\mathbb{P}(A_z|X=x)p_X(x)}{\mathbb{P}(A_z)} = \dfrac{p_z(x)p_X(x)}{p_z}$, we have

$$\begin{aligned}
F_z(t) &= \int F_z^x(t) p_{X|A_z}(x) dx \\
&= \frac{1}{p_z} \int F_z^x(t) p_z(x) p_X(x) dx \\
&= \frac{1}{p_z} \mathbb{E}[p_z(X) F_z^X(t)].
\end{aligned}$$

## B. Additional Experimental Details

### B.1. Training details

For CIFAR-100 models, we use the following: Batch size: 128; Epochs: 100; Cross entropy loss; Optimizer: SGD; Learning rate: 0.1; Momentum 0.9 and weight decay 0.0005.
Similarly, for Living 17 we use: Batch size: 64; Epochs: 15; Cross entropy loss; Optimizer: Adam; Learning rate: 0.0001.
For training details regarding Mini-ImageNet, see the following link:
https://huggingface.co/datasets/timm/mini-ImageNet

### B.2. Definitions of the score functions

**LAC**:

$$s(x,y) := 1 - \hat{\pi}(x,y) \tag{9}$$

**RAPS**:

$$s(x,y) := \sum_{y'=1}^{C} \hat{\pi}(x,y') \mathbf{1}\{\hat{\pi}(x,y') > \hat{\pi}(x,y)\} + \lambda_{RAPS} \cdot \left( o_x(y) - k_{\text{reg}} \right)^+ + \hat{\pi}(x,y) \cdot u, \tag{10}$$

where

$$o_x(y) = \left| \{ y' \in \mathcal{Y} : \hat{\pi}(x,y') \geq \hat{\pi}(x,y) \} \right|,$$

$(x)^+$ is the positive part of the expression and $\lambda_{RAPS}, k_{reg}$ are hyperparameters, which we set as in the original RAPS implementation.

**SAPS**:

$$S(x, y) := \begin{cases} u \cdot \hat{\pi}_{\max}(x, y), & \text{if } o_x(y) = 1 \\ \hat{\pi}_{\max}(x, y) + \left(o_x(y) - 2 + u\right) \cdot \lambda_{SAPS}, & \text{else} \end{cases} \tag{11}$$

where $u$ is a uniform random variable and $\hat{\pi}_{\max}(x, y)$ denotes the maximum softmax. We optimized the hyperparameter $\lambda_{SAPS}$ per model-dateset pair to a fixed value that minimizes the average set size (the values where roughly around 0.08).

### B.3. Definitions of the evaluation metrics

We report metrics over the test set, which we denote by $\{(\mathbf{x}_i^{(test)}, y_i^{(test)})\}_{i=1}^{N_{test}}$, comprising of the samples that were not included in the calibration set. The metrics are as follows.

- *Average set size* – The mean prediction set size of the CP algorithm:

$$\text{Average Size} = \frac{1}{N_{test}} \sum_{i=1}^{N_{test}} |\mathcal{C}(\mathbf{x}_i^{(test)})|.$$

- *Average number of superclasses* - The mean number of distinct superclasses in prediction set of the CP algorithm.

$$\text{Average \#Superclasses} = \frac{1}{N_{test}} \sum_{i=1}^{N_{test}} |\mathcal{G}(\mathbf{x}_i^{(test)})|.$$

where $\mathcal{G}(x) = \{g(y) : y \in \mathcal{C}(x)\}$

- *Marginal coverage* - The coverage rate of the prediction sets of the CP algorithm:

$$\text{Coverage rate} = \frac{1}{N_{test}} \sum_{i=1}^{N_{test}} \mathbf{1}\{y_i \in \mathcal{C}(x_i^{(test)})\}.$$

- *Worst class-conditional coverage gap* - The highest deviation from the desired 1- $\alpha$ coverage:

$$\text{TopCovGap} = \text{Max}_{y \in [C]} \left| \frac{1}{|I_y|} \sum_{i \in I_y} \mathbf{1}\left\{y_i^{(\text{test})} \in \mathcal{C}\left(x_i^{(\text{test})}\right)\right\} - (1 - \alpha) \right|,$$

where $I_y = \{i \in [N_{test}] : y_i^{(\text{test})} = y\}$ is the indices of the test examples labeled $y$. This metric is similar to those used in previous works (Ding et al., 2023; Dabah and Tirer, 2025).

## C. Additional Experiments

### C.1. Marginal coverage

In Table 3, we present the marginal coverage for each of the methods and all the settings for $\alpha = 0.05$. Similarly, in Table 4 we present the marginal coverage for $\alpha = 0.1$. As expected, the marginal coverage holds.

### C.2. Class-conditional coverage

In Table 5, we present the worst class-conditional coverage gap for each of the methods and all the settings for $\alpha = 0.05$. Similarly, in Table 6 we present the marginal coverage for $\alpha = 0.1$. It can been observed that overall our class-similarity penalty does not significantly change the metric compared to the standard versions. In fact, for LAC and SAPS our MS-CS often yields some improvement.

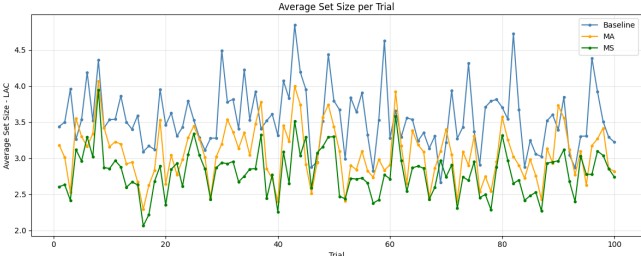

*Figure 4.* Average set size per trial using CIFAR100-ResNet50 with $\alpha = 0.05$ and LAC score. MA-CS outperforms the baseline in 91/100 trials and MS-CS outperforms the baseline in 98/100 trials.

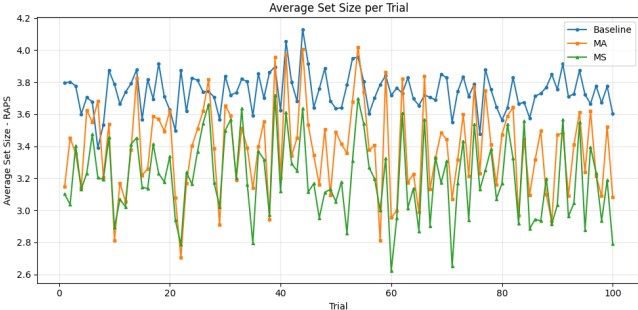

*Figure 5.* Average set size per trial using CIFAR100-ResNet50 with $\alpha = 0.05$ and RAPS score. MA-CS outperforms the baseline in 93/100 trials and MS-CS outperforms the baseline in 100/100 trials.

### C.3. Consistent improvement by our methods even when error bars overlap

We show that even in cases where there is overlap between the error bars (of $\pm$ standard deviation) in Table 1, our methods consistently outperform the baseline. To clarify these, we present per-trial comparisons over 100 runs for LAC in Figure 4 and for RAPS in Figure 5 for overlapping cases for CIFAR100-ResNet50 with $\alpha = 0.05$. These figures show that improvements are consistent rather than driven by outliers. For LAC, MA-CS outperforms the baseline in 91/100 trials and MS-CS outperforms the baseline in 98/100 trials. For RAPS, MA-CS outperforms the baseline in 93/100 trials and MS-CS outperforms the baseline in 100/100 trials. Thus, despite occasional overlap in error bars, the improvements are systematic and robust.

### C.4. Additional experimental configurations.

We conduct additional experiments such as varying the amount of calibration data and examining distribution shift at test time.

Table 7 reflects a small calibration regime (10% calibration / 90% test), where our methods consistently outperform the standard RAPS baseline. For example, on CIFAR-100 with ResNet-34, the average set size decreases from $5.77$ to $4.52$ (MA-CS) and $4.18$ (MS-CS), while the number of superclasses drops from $3.39$ to $\sim 2.28$–$2.36$.

We also evaluate robustness under mild distribution shift in Tables 8 and 9 (Gaussian noise, brightness; following (Hendrycks and Dietterich, 2019)) for the RAPS score. Under brightness mismatch with CIFAR-100 and ResNet-50, the set size improves from $3.36$ to $2.92$ (MA-CS) and $2.74$ (MS-CS), and the #Superclasses reduces from $2.37$ to $\sim 1.84$. Similar gains hold under Gaussian noise.

*Table 3.* Marginal coverage of the CP methods for $\alpha = 0.05$.

| Method | Coverage | | | | |
| --- | --- | --- | --- | --- | --- |
| | ImageNet, ViT | m-ImageNet, RN50 | CIFAR100, RN50 | CIFAR100, RN34 | L17, RN50 |
| **LAC** | | | | | |
| Standard | 0.950 ($\pm$0.002) | 0.952 ($\pm$0.002) | 0.950 ($\pm$0.001) | 0.952 ($\pm$0.003) | 0.953 ($\pm$0.002) |
| Clustered | 0.952 ($\pm$0.003) | 0.950 ($\pm$0.002) | 0.951 ($\pm$0.002) | 0.950 ($\pm$0.001) | 0.950 ($\pm$0.002) |
| AIR | N/A | N/A | 0.951 ($\pm$0.004) | 0.954 ($\pm$0.003) | 0.948 ($\pm$0.003) |
| MA-CS | N/A | N/A | 0.950 ($\pm$0.002) | 0.951 ($\pm$0.002) | 0.949 ($\pm$0.002) |
| MS-CS | 0.952 ($\pm$0.001) | 0.949 ($\pm$0.003) | 0.950 ($\pm$0.002) | 0.947 ($\pm$0.004) | 0.950 ($\pm$0.001) |
| **RAPS** | | | | | |
| Standard | 0.950 ($\pm$0.001) | 0.951 ($\pm$0.002) | 0.950 ($\pm$0.001) | 0.951 ($\pm$0.002) | 0.955 ($\pm$0.003) |
| Clustered | 0.953 ($\pm$0.002) | 0.950 ($\pm$0.002) | 0.950 ($\pm$0.003) | 0.951 ($\pm$0.001) | 0.951 ($\pm$0.002) |
| AIR | N/A | N/A | 0.950 ($\pm$0.002) | 0.952 ($\pm$0.002) | 0.951 ($\pm$0.003) |
| MA-CS | N/A | N/A | 0.950 ($\pm$0.001) | 0.949 ($\pm$0.002) | 0.952 ($\pm$0.002) |
| MS-CS | 0.950 ($\pm$0.002) | 0.951 ($\pm$0.003) | 0.948 ($\pm$0.002) | 0.948 ($\pm$0.003) | 0.950 ($\pm$0.001) |
| **SAPS** | | | | | |
| Standard | 0.948 ($\pm$0.003) | 0.952 ($\pm$0.002) | 0.950 ($\pm$0.002) | 0.951 ($\pm$0.001) | 0.950 ($\pm$0.002) |
| Clustered | 0.951 ($\pm$0.002) | 0.950 ($\pm$0.001) | 0.951 ($\pm$0.003) | 0.949 ($\pm$0.002) | 0.950 ($\pm$0.002) |
| AIR | N/A | N/A | 0.954 ($\pm$0.004) | 0.946 ($\pm$0.003) | 0.950 ($\pm$0.002) |
| MA-CS | N/A | N/A | 0.946 ($\pm$0.003) | 0.943 ($\pm$0.005) | 0.951 ($\pm$0.002) |
| MS-CS | 0.952 ($\pm$0.002) | 0.948 ($\pm$0.002) | 0.946 ($\pm$0.004) | 0.949 ($\pm$0.003) | 0.951 ($\pm$0.002) |

*Table 4.* Marginal coverage of the CP methods for $\alpha = 0.1$.

| Method | Coverage | | | | |
| --- | --- | --- | --- | --- | --- |
| | ImageNet, ViT | m-ImageNet, RN50 | CIFAR100, RN50 | CIFAR100, RN34 | L17, RN50 |
| **LAC** | | | | | |
| Standard | 0.905 ($\pm$0.002) | 0.901 ($\pm$0.002) | 0.899 ($\pm$0.003) | 0.902 ($\pm$0.002) | 0.905 ($\pm$0.001) |
| Clustered | 0.919 ($\pm$0.004) | 0.895 ($\pm$0.003) | 0.915 ($\pm$0.004) | 0.901 ($\pm$0.002) | 0.902 ($\pm$0.002) |
| AIR | N/A | N/A | 0.905 ($\pm$0.002) | 0.902 ($\pm$0.002) | 0.901 ($\pm$0.001) |
| MA-CS | N/A | N/A | 0.900 ($\pm$0.001) | 0.895 ($\pm$0.003) | 0.898 ($\pm$0.002) |
| MS-CS | 0.899 ($\pm$0.002) | 0.896 ($\pm$0.002) | 0.897 ($\pm$0.002) | 0.902 ($\pm$0.001) | 0.899 ($\pm$0.001) |
| **RAPS** | | | | | |
| Standard | 0.903 ($\pm$0.002) | 0.898 ($\pm$0.003) | 0.900 ($\pm$0.002) | 0.905 ($\pm$0.002) | 0.902 ($\pm$0.002) |
| Clustered | 0.899 ($\pm$0.003) | 0.902 ($\pm$0.002) | 0.917 ($\pm$0.004) | 0.913 ($\pm$0.003) | 0.903 ($\pm$0.002) |
| AIR | N/A | N/A | 0.906 ($\pm$0.002) | 0.907 ($\pm$0.002) | 0.904 ($\pm$0.002) |
| MA-CS | N/A | N/A | 0.899 ($\pm$0.001) | 0.899 ($\pm$0.001) | 0.900 ($\pm$0.001) |
| MS-CS | 0.903 ($\pm$0.002) | 0.901 ($\pm$0.002) | 0.898 ($\pm$0.003) | 0.900 ($\pm$0.002) | 0.905 ($\pm$0.002) |
| **SAPS** | | | | | |
| Standard | 0.899 ($\pm$0.002) | 0.900 ($\pm$0.001) | 0.898 ($\pm$0.002) | 0.904 ($\pm$0.003) | 0.899 ($\pm$0.002) |
| Clustered | 0.904 ($\pm$0.002) | 0.901 ($\pm$0.002) | 0.900 ($\pm$0.001) | 0.900 ($\pm$0.001) | 0.902 ($\pm$0.002) |
| AIR | N/A | N/A | 0.908 ($\pm$0.003) | 0.896 ($\pm$0.002) | 0.900 ($\pm$0.001) |
| MA-CS | N/A | N/A | 0.898 ($\pm$0.002) | 0.900 ($\pm$0.001) | 0.898 ($\pm$0.002) |
| MS-CS | 0.901 ($\pm$0.001) | 0.898 ($\pm$0.002) | 0.900 ($\pm$0.002) | 0.899 ($\pm$0.001) | 0.900 ($\pm$0.001) |

*Table 5.* Worst class-conditional coverage gap (TopCovGap) of the CP methods for $\alpha = 0.05$.

| Method | ImageNet, ViT | m-ImageNet, RN50 | CIFAR100, RN50 | CIFAR100, RN34 | L17, RN50 |
|---|---|---|---|---|---|
| | | | TopCovGap | | |
| **LAC** | | | | | |
| Standard | 0.265 ($\pm$0.038) | 0.106 ($\pm$0.015) | 0.101 ($\pm$0.012) | 0.114 ($\pm$0.018) | 0.182 ($\pm$0.024) |
| Clustered | 0.300 ($\pm$0.044) | 0.125 ($\pm$0.032) | 0.102 ($\pm$0.028) | 0.118 ($\pm$0.035) | 0.226 ($\pm$0.041) |
| AIR | N/A | N/A | 0.086 ($\pm$0.011) | 0.123 ($\pm$0.019) | 0.127 ($\pm$0.020) |
| MA-CS | N/A | N/A | 0.111 ($\pm$0.014) | 0.125 ($\pm$0.016) | 0.206 ($\pm$0.029) |
| MS-CS | 0.281 ($\pm$0.035) | 0.128 ($\pm$0.021) | 0.096 ($\pm$0.013) | 0.109 ($\pm$0.015) | 0.168 ($\pm$0.022) |
| **RAPS** | | | | | |
| Standard | 0.208 ($\pm$0.030) | 0.122 ($\pm$0.018) | 0.078 ($\pm$0.010) | 0.083 ($\pm$0.011) | 0.160 ($\pm$0.021) |
| Clustered | 0.255 ($\pm$0.040) | 0.143 ($\pm$0.038) | 0.099 ($\pm$0.025) | 0.092 ($\pm$0.029) | 0.183 ($\pm$0.033) |
| AIR | N/A | N/A | 0.061 ($\pm$0.009) | 0.093 ($\pm$0.012) | 0.097 ($\pm$0.014) |
| MA-CS | N/A | N/A | 0.099 ($\pm$0.013) | 0.135 ($\pm$0.017) | 0.169 ($\pm$0.023) |
| MS-CS | 0.296 ($\pm$0.039) | 0.136 ($\pm$0.022) | 0.090 ($\pm$0.014) | 0.118 ($\pm$0.019) | 0.160 ($\pm$0.025) |
| **SAPS** | | | | | |
| Standard | 0.243 ($\pm$0.032) | 0.128 ($\pm$0.017) | 0.106 ($\pm$0.015) | 0.134 ($\pm$0.020) | 0.213 ($\pm$0.028) |
| Clustered | 0.295 ($\pm$0.042) | 0.136 ($\pm$0.036) | 0.091 ($\pm$0.027) | 0.145 ($\pm$0.039) | 0.218 ($\pm$0.045) |
| AIR | N/A | N/A | 0.070 ($\pm$0.010) | 0.099 ($\pm$0.013) | 0.100 ($\pm$0.014) |
| MA-CS | N/A | N/A | 0.118 ($\pm$0.016) | 0.173 ($\pm$0.024) | 0.186 ($\pm$0.026) |
| MS-CS | 0.259 ($\pm$0.034) | 0.144 ($\pm$0.020) | 0.095 ($\pm$0.014) | 0.115 ($\pm$0.018) | 0.219 ($\pm$0.030) |

*Table 6.* Worst class-conditional coverage gap (TopCovGap) of the CP methods for $\alpha = 0.1$.

| Method | ImageNet, ViT | m-ImageNet, RN50 | CIFAR100, RN50 | CIFAR100, RN34 | L17, RN50 |
|---|---|---|---|---|---|
| | | | TopCovGap | | |
| **LAC** | | | | | |
| Standard | 0.443 ($\pm$0.051) | 0.180 ($\pm$0.022) | 0.194 ($\pm$0.024) | 0.169 ($\pm$0.020) | 0.372 ($\pm$0.045) |
| Clustered | 0.370 ($\pm$0.056) | 0.183 ($\pm$0.026) | 0.186 ($\pm$0.025) | 0.179 ($\pm$0.023) | 0.312 ($\pm$0.039) |
| AIR | N/A | N/A | 0.196 ($\pm$0.027) | 0.211 ($\pm$0.029) | 0.306 ($\pm$0.042) |
| MA-CS | N/A | N/A | 0.143 ($\pm$0.018) | 0.183 ($\pm$0.021) | 0.372 ($\pm$0.048) |
| MS-CS | 0.403 ($\pm$0.049) | 0.183 ($\pm$0.023) | 0.164 ($\pm$0.020) | 0.192 ($\pm$0.025) | 0.346 ($\pm$0.041) |
| **RAPS** | | | | | |
| Standard | 0.297 ($\pm$0.035) | 0.129 ($\pm$0.015) | 0.139 ($\pm$0.017) | 0.101 ($\pm$0.012) | 0.224 ($\pm$0.028) |
| Clustered | 0.316 ($\pm$0.048) | 0.134 ($\pm$0.019) | 0.149 ($\pm$0.020) | 0.140 ($\pm$0.018) | 0.235 ($\pm$0.030) |
| AIR | N/A | N/A | 0.144 ($\pm$0.019) | 0.120 ($\pm$0.016) | 0.102 ($\pm$0.014) |
| MA-CS | N/A | N/A | 0.131 ($\pm$0.017) | 0.176 ($\pm$0.024) | 0.261 ($\pm$0.034) |
| MS-CS | 0.411 ($\pm$0.052) | 0.163 ($\pm$0.022) | 0.148 ($\pm$0.018) | 0.177 ($\pm$0.023) | 0.277 ($\pm$0.035) |
| **SAPS** | | | | | |
| Standard | 0.258 ($\pm$0.032) | 0.188 ($\pm$0.024) | 0.156 ($\pm$0.021) | 0.180 ($\pm$0.023) | 0.368 ($\pm$0.046) |
| Clustered | 0.322 ($\pm$0.046) | 0.180 ($\pm$0.025) | 0.175 ($\pm$0.022) | 0.181 ($\pm$0.024) | 0.302 ($\pm$0.040) |
| AIR | N/A | N/A | 0.146 ($\pm$0.018) | 0.251 ($\pm$0.034) | 0.198 ($\pm$0.026) |
| MA-CS | N/A | N/A | 0.135 ($\pm$0.016) | 0.201 ($\pm$0.025) | 0.346 ($\pm$0.047) |
| MS-CS | 0.284 ($\pm$0.038) | 0.197 ($\pm$0.027) | 0.158 ($\pm$0.020) | 0.190 ($\pm$0.024) | 0.273 ($\pm$0.036) |

*Table 7.* Performance comparison of various CP methods under RAPS score with $\alpha = 0.05$ for (10%, 90%) split.

| | #Superclasses ↓ | | Size ↓ | | | |
|---|---|---|---|---|---|---|
| Method | CIFAR100, RN50 | CIFAR100, RN34 | ImageNet, ViT | m-ImageNet, RN50 | CIFAR100, RN50 | CIFAR100, RN34 |
| Standard | 2.44 ($\pm$0.107) | 3.39 ($\pm$0.122) | 3.29 ($\pm$0.175) | 8.40 ($\pm$1.915) | 3.73 ($\pm$0.206) | 5.77 ($\pm$0.246) |
| Clustered | 2.41 ($\pm$0.111) | 3.31 ($\pm$0.135) | 3.17 ($\pm$0.062) | 8.35 ($\pm$2.156) | 3.65 ($\pm$0.214) | 5.60 ($\pm$0.274) |
| AIR | **1.96** ($\pm$0.079) | 3.01 ($\pm$0.080) | N/A | N/A | 9.8 ($\pm$0.124) | 15.05 ($\pm$0.092) |
| MA-CS | 2.01 ($\pm$0.223) | **2.28** ($\pm$0.323) | N/A | N/A | 3.64 ($\pm$0.582) | 4.52 ($\pm$0.547) |
| MS-CS | 2.02 ($\pm$0.201) | 2.36 ($\pm$0.261) | **2.54** ($\pm$0.382) | **8.07** ($\pm$1.942) | **3.32** ($\pm$0.414) | **4.18** ($\pm$0.593) |

*Table 8.* Performance comparison of various CP methods under RAPS score with $\alpha = 0.1$ for brightness mismatch.

| | #Superclasses ↓ | | Size ↓ | | |
|---|---|---|---|---|---|
| Method | CIFAR100, RN50 | CIFAR100, RN34 | ImageNet, ViT | CIFAR100, RN50 | CIFAR100, RN34 |
| Standard | 2.37 ($\pm$0.058) | 1.95 ($\pm$0.061) | 2.44 ($\pm$0.104) | 3.36 ($\pm$0.108) | 2.58 ($\pm$0.106) |
| MA-CS | **1.83** ($\pm$0.154) | **1.60** ($\pm$0.119) | N/A | 2.92 ($\pm$0.253) | 2.36 ($\pm$0.186) |
| MS-CS | 1.85 ($\pm$0.133) | 1.61 ($\pm$0.166) | **1.30** ($\pm$0.077) | **2.74** ($\pm$0.242) | **2.22** ($\pm$0.087) |

*Table 9.* Performance comparison of various CP methods under RAPS score with $\alpha = 0.1$, Gaussian noise.

| | #Superclasses ↓ | | Size ↓ | | |
|---|---|---|---|---|---|
| Method | CIFAR100, RN50 | CIFAR100, RN34 | ImageNet, ViT | CIFAR100, RN50 | CIFAR100, RN34 |
| Standard | 3.52 ($\pm$0.134) | 3.76 ($\pm$0.446) | 2.45 ($\pm$0.124) | 5.25 ($\pm$0.240) | 5.48 ($\pm$0.737) |
| MA-CS | 3.11 ($\pm$0.190) | **3.13** ($\pm$0.506) | N/A | 5.05 ($\pm$0.438) | 5.29 ($\pm$0.847) |
| MS-CS | **3.10** ($\pm$0.279) | 3.29 ($\pm$0.513) | **1.31** ($\pm$0.096) | **4.83** ($\pm$0.483) | **5.16** ($\pm$0.895) |

# D. Visualization and Ablation Study of the Model-Specific Class Similarity

## D.1. Visualization

In this section, we examine the cosine class similarity, as described in Section 5, using the class similarity matrix $M$. The image on the left shows the full similarity matrix, in contrast to the 10×10 zoomed-in view presented in the left panel of Figure 3. The right image similarly displays the full, unzoomed version of the model-agnostic superclass association matrix.

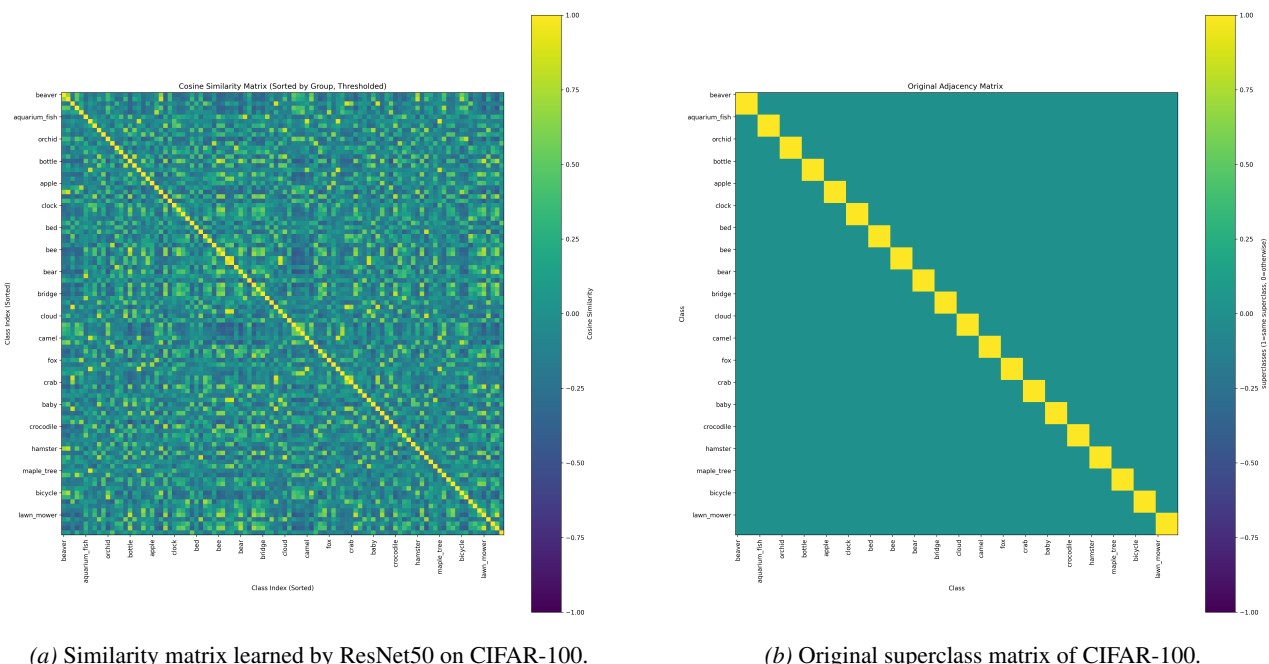

*(a)* Similarity matrix learned by ResNet50 on CIFAR-100.                *(b)* Original superclass matrix of CIFAR-100.

*Figure 6.* Comparison of the ResNet50 model-specific similarity matrix and the original superclass matrix of CIFAR-100.

## D.2. Ablation study

In this section, we further emphasize the benefits of incorporating model-perceived class similarity. To this end, we compare the performance of our MS-CS matrix, which uses a similarity matrix $M$ detailed in Section 5, with a version that uses a simple identity matrix $M = I$, which is model-agnostic and penalizes all non-predicted classes equally. We refer to this baseline as Model-Agnostic Diagonal (MA-Diag). In addition, we add another version of model agnostic which uses random (incorrect) similarity between classes (the diagonal of $M$ is always included). We refer to it as MA-Random. Tuning the hyperparameter $\lambda$ for MA-Diag and MA-Random is done exactly as for MA-CS and MS-CS.

Tables 10 and 11 report the resulting number of superclasses and prediction set sizes across all dataset–model pairs and CP algorithms for $\alpha \in \{0.05, 0.1\}$, respectively. Across all settings and for both evaluation metrics, MA-Diag and MA-Random never outperform either MA-CS or MS-CS. This demonstrates the clear advantage of accounting for inter-class similarity—whether derived from semantic structure or captured implicitly by the model's learned embeddings, and the quality of the similarity. We also note that, unlike our method, the naive MA-Diag does not reduce the average prediction set size compared to the standard LAC (cf. Tables 1 and 2).

*Table 10.* Performance comparison of model agnostic and specific methods with $\alpha = 0.05$. MA-Diag and MA-Random are ablated variants.

| Method | #Superclasses ↓ | | | Size ↓ | | | |
|---|---|---|---|---|---|---|---|
| | CIFAR100, RN50 | CIFAR100, RN34 | L17, RN50 | Mini-ImageNet | CIFAR100, RN50 | CIFAR100, RN34 | L17, RN50 |
| **LAC** | | | | | | | |
| MA-CS | 1.85 ($\pm$0.183) | 1.92 ($\pm$0.108) | **1.19** ($\pm$0.068) | N/A | 3.17 ($\pm$0.424) | 3.51 ($\pm$0.749) | 1.71 ($\pm$0.183) |
| MS-CS | **1.83** ($\pm$0.137) | **1.87** ($\pm$0.126) | **1.19** ($\pm$0.058) | **3.82** ($\pm$0.696) | **2.92** ($\pm$0.339) | **2.94** ($\pm$0.339) | **1.70** ($\pm$0.156) |
| MA-Diag | 2.29 ($\pm$0.285) | 2.38 ($\pm$0.285) | 1.26 ($\pm$0.059) | 4.93 ($\pm$1.050) | 3.74 ($\pm$0.785) | 3.74 ($\pm$0.724) | 1.77 ($\pm$0.177) |
| MA-Random | 5.58 ($\pm$0.202) | 5.26 ($\pm$1.074) | 1.25 ($\pm$0.065) | N/A | 7.45 ($\pm$0.640) | 7.08 ($\pm$1.477) | 1.74 ($\pm$0.185) |
| **RAPS** | | | | | | | |
| MA-CS | 2.01 ($\pm$0.160) | 2.29 ($\pm$0.250) | **1.23** ($\pm$0.081) | N/A | 3.50 ($\pm$0.305) | 4.52 ($\pm$0.501) | 1.97 ($\pm$0.295) |
| MS-CS | **1.95** ($\pm$0.122) | **2.22** ($\pm$0.182) | 1.24 ($\pm$0.050) | **7.35** ($\pm$2.495) | **3.17** ($\pm$0.265) | **3.79** ($\pm$0.387) | **1.81** ($\pm$0.222) |
| MA-Diag | 2.40 ($\pm$0.144) | 3.13 ($\pm$0.186) | 1.32 ($\pm$0.071) | 8.63 ($\pm$2.256) | 3.65 ($\pm$0.282) | 5.23 ($\pm$0.389) | 1.87 ($\pm$0.236) |
| MA-Random | 2.45 ($\pm$0.121) | 3.18 ($\pm$0.179) | 1.38 ($\pm$0.095) | N/A | 3.73 ($\pm$0.231) | 5.29 ($\pm$0.377) | 1.98 ($\pm$0.270) |
| **SAPS** | | | | | | | |
| MA-CS | **1.88** ($\pm$0.286) | **1.93** ($\pm$0.162) | 1.26 ($\pm$0.033) | N/A | **3.14** ($\pm$0.464) | **3.32** ($\pm$0.271) | 1.94 ($\pm$0.147) |
| MS-CS | 1.97 ($\pm$0.187) | 2.14 ($\pm$0.175) | **1.24** ($\pm$0.035) | **4.7** ($\pm$1.026) | **3.14** ($\pm$0.361) | **3.32** ($\pm$0.371) | **1.87** ($\pm$0.146) |
| MA-Diag | 2.29 ($\pm$0.203) | 2.36 ($\pm$0.176) | 1.32 ($\pm$0.105) | 5.61 ($\pm$1.441) | 3.38 ($\pm$0.393) | 3.52 ($\pm$0.346) | 1.99 ($\pm$0.357) |
| MA-Random | 2.31 ($\pm$0.191) | 2.38 ($\pm$0.145) | 1.34 ($\pm$0.112) | N/A | 3.43 ($\pm$0.373) | 3.55 ($\pm$0.289) | 2.02 ($\pm$0.352) |

*Table 11.* Performance comparison of model agnostic and specific methods with $\alpha = 0.1$. MA-Diag and MA-Random are ablated variants.

| Method | #Superclasses ↓ | | | Size ↓ | | | |
|---|---|---|---|---|---|---|---|
| | CIFAR100, RN50 | CIFAR100, RN34 | L17, RN50 | Mini-ImageNet | CIFAR100, RN50 | CIFAR100, RN34 | L17, RN50 |
| **LAC** | | | | | | | |
| MA-CS | **1.24** ($\pm$0.063) | 1.34 ($\pm$0.065) | **1.04** ($\pm$0.018) | N/A | 1.54 ($\pm$0.127) | 1.70 ($\pm$0.120) | 1.19 ($\pm$0.037) |
| MS-CS | 1.25 ($\pm$0.053) | **1.32** ($\pm$0.072) | 1.05 ($\pm$0.015) | **1.69** ($\pm$0.158) | **1.53** ($\pm$0.092) | **1.65** ($\pm$0.138) | **1.18** ($\pm$0.042) |
| MA-Diag | 1.39 ($\pm$0.067) | 1.44 ($\pm$0.068) | 1.08 ($\pm$0.040) | 1.79 ($\pm$0.209) | 1.67 ($\pm$0.123) | 1.74 ($\pm$0.121) | 1.25 ($\pm$0.113) |
| MA-Random | 1.39 ($\pm$0.120) | 1.34 ($\pm$0.065) | 1.07 ($\pm$0.030) | N/A | 1.65 ($\pm$0.189) | 1.70 ($\pm$0.124) | 1.21 ($\pm$0.086) |
| **RAPS** | | | | | | | |
| MA-CS | **1.34** ($\pm$0.099) | **1.45** ($\pm$0.100) | **1.03** ($\pm$0.032) | N/A | 2.10 ($\pm$0.177) | 2.60 ($\pm$0.165) | 1.38 ($\pm$0.053) |
| MS-CS | **1.34** ($\pm$0.085) | 1.49 ($\pm$0.080) | 1.07 ($\pm$0.020) | **2.05** ($\pm$0.203) | **1.89** ($\pm$0.160) | **2.18** ($\pm$0.174) | **1.28** ($\pm$0.056) |
| MA-Diag | 1.67 ($\pm$0.099) | 1.99 ($\pm$0.115) | 1.14 ($\pm$0.036) | 2.29 ($\pm$0.212) | 2.19 ($\pm$0.199) | 2.82 ($\pm$0.241) | 1.31 ($\pm$0.091) |
| MA-Random | 1.73 ($\pm$0.103) | 2.11 ($\pm$0.131) | 1.18($\pm$0.048) | N/A | 2.29 ($\pm$0.205) | 2.99 ($\pm$0.273) | 1.41 ($\pm$0.107) |
| **SAPS** | | | | | | | |
| MA-CS | **1.26** ($\pm$0.044) | 1.42 ($\pm$0.063) | **1.06** ($\pm$0.012) | N/A | 1.74 ($\pm$0.140) | 1.89 ($\pm$0.163) | 1.27 ($\pm$0.062) |
| MS-CS | 1.36 ($\pm$0.084) | **1.39** ($\pm$0.059) | 1.07 ($\pm$0.013) | **1.95** ($\pm$0.239) | **1.71** ($\pm$0.172) | **1.81** ($\pm$0.136) | **1.24** ($\pm$0.049) |
| MA-Diag | 1.45 ($\pm$0.094) | 1.54 ($\pm$0.093) | 1.10 ($\pm$0.034) | 2.07 ($\pm$0.330) | 1.78 ($\pm$0.179) | 1.92 ($\pm$0.182) | 1.27 ($\pm$0.105) |
| MA-Random | 1.45 ($\pm$0.089) | 1.53 ($\pm$0.086) | 1.10($\pm$0.042) | N/A | 1.77 ($\pm$0.165) | 1.90 ($\pm$0.165) | 1.29 ($\pm$0.127) |

