# OpenReview forum: "Enhancing Conformal Prediction via Class Similarity"
_ICML.cc/2026/Conference — ICML 2026 regular_

### Official Review · Reviewer_oqAa · 2026-03-09

**Soundness:** 4
**Presentation:** 3
**Significance:** 4
**Originality:** 4
**Overall Recommendation:** 6
**Confidence:** 3

**Summary:**

The paper modifies standard conformity scores to make them more efficient in the situation where the classes are grouped into semantically meaningful superclasses. The approach is then extended to automatic groupings of classes. Extensive empirical studies are reported.

When referring to line numbers in the paper, I will use L/R to mean the left/right column.

**Compliance With Llm Reviewing Policy:**

Affirmed.

**Final Justification:**

The authors have addressed all my questions and concerns reinforcing my prior assessment.

**Key Questions For Authors:**

1 A natural approach to predicting classes grouped into superclasses would be first to apply your modification of CP to superclasses in place of classes, and as the next step to refine the predicted superclasses by predicting individual classes. Does this approach have obvious disadvantages as compared to what you are doing? Or does AIR do something similar?

2 In Assumption 1, the prediction set $C_{\lambda}(X)$ is based on the true statistical quantile $q_{\lambda}$. How is it possible? The true distribution is not observable. If it's impossible (which the authors' discussion after the assumptions seems to imply), my suggestion is to use an expression such as "ideal prediction set".

Some less important questions are asked in other sections of the review.

**Limitations:**

The authors only penalize superclasses different from the superclass containing $\hat y(x)$, and their penalty term does not encourage minimizing the number of other superclasses. Can it be the reason for the better performance of the AIR in some cases?

**Strengths And Weaknesses:**

This paper proposes methods that improve greatly the efficiency of CP in the situation where classes are grouped into superclasses; and these methods continue to work even when no such grouping is given.

Proposition 4.2 is a simple but very nice observation.

Results of the empirical studies reported in Section 6 look great, and the methods proposed in this paper consistently outperform
the existing methods. The strongest competitor is AIR motivated by Goren et al.'s Climbing Inference Rule. AIR is an improved variant developed by the authors, and it would be great to have the details of the algorithm (at least in the appendix); at the moment the description is very brief.

Minor comments about presentation:
* Line 135L: when you say "post-softmax probability vector", do you mean that the components of $\hat\pi(x)$ are nonnegative and sum to 1? It would be clearer.
* Line 136.5L: the expression $\hat y(x) = \arg\max_i \hat\pi_i(x)$ is not rigorous; the max can be attained at several $y$.
    How do you define $\hat y(x)$ in this case? Or do your assumptions prevent ties?
* Line 115R: it should be mentioned that the lower bound requires some assumptions (such as all scores being different almost surely).
* Line 197.5L: when you write $G_{\lambda}(x) \subseteq G(x)$, do you mean $\mathcal{G}_{\lambda}(x) \subseteq \mathcal{G}(x)$?
* Line 373R: it appears that you mean CIFAR100-ResNet34 when you write "CIFAR100-ResNet50". (The advantage of your methods over AIR for CIFAR100-ResNet50 is also significant.)

---

> ### Author Rebuttal · Authors · 2026-03-30
>
> We thank the reviewer for their positive feedback and are pleased from their evaluation of our work.
> We address their concerns below.
>
> **Q1 (Applying CP to superclasses in place of classes):**
>
> The competing method AIR works by applying CP on superclasses rather than on individual classes.
> Specifically, it operates by sequentially accumulating superclasses to preserve coverage (improving over exiting hierarchical methods like (Goren et al., 2024) that climb the superclass tree). In the appendix of the revision we will present AIR formally in an algorithm box.
> Our results show clear efficiency disadvantages (larger sets) of this approach compared to our proposed method.
>
> The reviewer's 2-steps suggestion of applying CP on superclasses and after this applying class-resolution refinement is interesting, but it is not clear to us how it can be done while preserving the desired theoretical properties of CP.
> In contrast, the success of our approach may come from directly considering class-level scores and superclass-level penalty in a single step.
>
> **Q2 (Assumption 1 of using the statistical quantile $q_{\lambda}$):**
>
> As discussed below it, Assumption 1 essentially reflects having an infinite calibration set, which as stated, was made "for making the analysis
> tractable" by "sparing cumbersome analysis of the effect of finite calibration sets on inclusion of a label in the predictions sets."
>
> Note that the empirical score quantile $\\hat{q}\_{\\lambda}$ converges to statistical quantile $q\_{\\lambda}$ as the number of calibration samples increases. It does not require knowing the true distribution and does not depend on the quality of the score or the accuracy of the model (thus, the suggested expression "ideal prediction set" may not be clear enough).
>
> In the revision, we will further clarify the necessity of this assumption for proving Theorem 4.5.
> Importantly, note that our experiments show that the implications of Theorem 4.5 hold in practical setups with rather small calibration sets (following common benchmarks). Furthermore, for theoretical analyses of CP beyond marginal coverage, it is quite common to make the asymptotic assumption on the calibration set size
> and there exist also works with much stronger assumptions, such as Bayes optimal (oracle) classifiers [R1].
>
> [R1] Angelopoulos, Anastasios N., Rina Foygel Barber, and Stephen Bates. "Theoretical foundations of conformal prediction." arXiv preprint arXiv:2411.11824 (2024).
>
> **Limitation (Our method vs. AIR).**
>
> Note that our approach consistently yields significantly smaller prediction sets than AIR (the efficiency of AIR is poor also compared to the baseline).
> In terms of \#superclasses, the better performance of AIR in some cases can be explained by the fact that it applies CP directly on superclasses while our approach combines class-level scores with superclass-level penalty rather than purely focusing on superclasses.
>
> **Minor comments:**
>
> All the minor comments will be handled in the final version.
>
> * Line 135L:
> Indeed, by "post-softmax probability vector" we mean the softmax output is a vector whose components are nonnegative and sum to 1. This will be clarified.
>
> * Line 136.5L:
> Regarding $\arg\max_i \hat\pi_i(x)$,  we will clarify that we break ties by selecting the smallest index among maximizers. Note though that since the logits values are continuous, ties occur with probability zero.
>
> * Line 115R:
> The upper bound indeed additionally requires all scores being different almost surely, this will be added to the sentence to make it more precise.
>
> * Line 197.5L:
> Thank you for catching this typo. Yes, we intended to write $\mathcal{G}\_{\lambda}(x) \subseteq \mathcal{G}(x)$ and not $G\_{\lambda}(x) \subseteq G(x)$.
>
> * Line 373R:
> You are correct—the reported numbers correspond to CIFAR100-ResNet34 rather than CIFAR100-ResNet50. This will be fixed. As you noted, the advantage of our methods is significant in both cases.

---

> > ### Author Rebuttal · Reviewer_oqAa · 2026-03-31
> >
> > Thank you very much for your helpful answers.

---

> > > ### Author Response · Authors · 2026-04-01
> > >
> > > Dear Reviewer oqAa, we thank you again for your positive and thoughtful review of our work.
> > > We are pleased that you are satisfied with our response.

---

### Official Review · Reviewer_rkJU · 2026-03-12

**Soundness:** 3
**Presentation:** 3
**Significance:** 4
**Originality:** 3
**Overall Recommendation:** 5
**Confidence:** 3

**Summary:**

This study enhances standard conformal prediction methods with semantic class similarity information. The result is a simple yet effective penalization term added to the score function, with theoretical guarantees to reduce expected prediction set size under specific conditions. The penalization is extended to settings where class similarity is learned from data, although the theoretical guarantees do not apply here. Extensive empirical evaluation on image classification benchmarks, both with and without an oracle hierarchy of class labels, demonstrate the effectiveness of the proposed approach.

**Compliance With Llm Reviewing Policy:**

Affirmed.

**Final Justification:**

My recommendation is a 5 (Accept). The novel penalization term added to the CP score function is a simple and effective extension to account for class similarity between labels, and the theoretical discourse makes sense to me. The extension to learned class similarity (MS-CS) seems particularly useful in practice. Generally, the work is well-written and easy to follow.

My concerns were addressed in the rebuttal and I raised my score from 4 to 5. Although I still think the theory is somewhat limited in scope (Thm 4.5 is local at $\lambda=0$, no theory for MS-CS) compared to what is shown in the experiments, I agree with the authors' rebuttal that it is fair to defer such theoretical extensions to future work.

**Key Questions For Authors:**

1. Theorem 4.5 seems somewhat unnecessarily complicated in its appeal to _continuous_ score CDFs and a _differentiable_ expected prediction set size in order to show the new penalized score function could reduce prediction set size under certain conditions. Could the authors help clarify whether a simpler approach is possible, and if so, to implement it?
2. Could the authors help clarify the scope of Theorem 4.5, i.e. for what values of $\lambda$ it does or does not hold?
3. In the comparison between MA-CS and MS-CS, it is observed that MS is slightly better in some cases. However, I understand the difference between the methods is that MA uses oracle class structure information while MS learns this from data. So, how would MS be able to consistently outperform the oracle baseline? Furthermore, what is meant with “the use of smaller groups” in MS?

**Limitations:**

Yes

**Strengths And Weaknesses:**

**Justification of the overall recommendation**
* The novel penalization term added to the score function is an elegant extension to account for class similarity, and the theoretical discourse mostly makes sense to me, with some comments on the approach mentioned below. The extension to learned class similarity seems particularly useful for practical applications.
* This submission could be a valuable contribution, and I see potential to increase my overall recommendation upon adequate follow-up on my comments below.

**Justification of confidence**
* While I am comfortable with conformal prediction, I have limited expertise with regard to some of the related works, such that I cannot adequately assess this submission's positioning.

**Strengths**
1. Taking into account the semantic similarity between labels to reduce prediction set size of conformal methods represents a meaningful extension of known methods.
2. The modified score function is a simple and effective extension. The extension to learned class similarity seems particularly meaningful for practical applications.
3. This work takes an interesting and possibly novel approach to modify the softmax ranking based on an oracle or learned semantic group structure, although I do not have sufficient expertise on the related works to adequately assess the technical novelty of such a modification.
4. This work is generally well-written and well-structured, and (most of) the theoretical contributions make sense to me, with some comments below.

**Weaknesses**
1. **Theorem 4.5 formalizes an interesting phenomenon, but it seems somewhat unnecessarily complicated in its approach.**
* Theorem 4.5 necessitates an infinite calibration set, which could probably be relaxed if not appealing to _continuous_ CDFs and a _differentiable_ expected prediction set size.
* In informal terms, I understand the phenomenon at hand as the following, please correct me if I am wrong. For a positive change $\Delta\lambda>0$, this increases the CP threshold $\widehat{q}\_\lambda$ with at most $\Delta\lambda$. This removes some out-of-group labels from $\mathcal{C}\_\lambda$ that fall below the old threshold, but when penalized do not fall below the new threshold. Similarly, some in-group labels are now included in $\mathcal{C}\_\lambda$, because they are not penalized and fall in between the old and the new threshold. This intuition seems to align with the explanation above Def. 4.4. If the first effect is bigger than the second, we observe a decrease in prediction set size.
* My interpretation is that none of the above reasoning requires a continuity argument, nor an infinite calibration set. It would be helpful to clarify whether a simpler approach is possible, and if so, to implement it.

2. **It is unclear for what $\lambda$ Theorem 4.5 holds and how to interpret it.**
* Assumption 1 refers to “small $\lambda\geq 0$”, for which the magnitude is not further specified, and upon closer inspection of Theorem 4.5, this relates to the derivative of the expected prediction set size evaluated at $\lambda=0$. A clear specification for what $\lambda$ this theorem holds would be appreciated, and for which $\lambda$ it does not.
* Closely related, the intuition described above Def. 4.4 is not restricted to $\lambda=0$, as is currently the scope of Theorem 4.5, hence the scope could possibly be broadened. It would be interesting to investigate whether this is possible.
* My understanding of the takeaway of Theorem 4.5 is that increasing $\lambda$ _could_ have a beneficial effect on prediction set size under specific conditions, such as specific values of $\lambda$ and softmax vectors $\hat{\pi}(x)$. This is context-dependent and does not necessarily pertain to _most_ or _all_ circumstances, as the summary of the main contributions seems to suggest. A relaxation of these claims might be more appropriate to describe the phenomenon and formalizations at hand, but I am open to discuss the interpretation of the authors.

3. **Learning class similarity seems to complicate theory, not the continuous relaxation.**
* The continuous generalisation of binary class similarity makes sense to me, but it seems that going from oracle to unknown class similarity is the part which complicates the theoretical considerations. It would be helpful to make this distinction clear, and if possible extend the scope of theoretical results to account for oracle continuous class similarity.

4. **Experimental comparison between MA-CS and MS-CS.**
* In the experiments section, in the comparison between MA-CS and MS-CS, I have some reservations on the explanations given. It is observed that MS is slightly better in some cases, yet the difference between the methods is that MA uses oracle group information, so it is unclear to me how this could lead to detrimental results.

5. **Notes on clarity of presentation and notation.**
* The notation $\mathcal{C}_\alpha$ and $\mathcal{C}$ is used interchangeably in various places, e.g. when defined on p.3 and in Theorem 3.1. Consistent use of one or the other seems more appropriate.
* The experiments talk about ‘superclasses’, and these seem equivalent to the ‘groups’ in earlier sections, but consistent use of one or the other seems more appropriate.
* Table 2 might be more appropriately placed in the Appendix, when Table 1 is already included in the main text and the findings are similar. Perhaps the leftover space could be used to enlarge Figure 2 and 3, since these figures and the corresponding axis ticks and labels appear small.
* A minor style issue, but the grey std in all tables are almost invisible to my weak eyes.
* It is mentioned that “AIR and our MA-CS cannot be applied for ImageNet and Mini-ImageNet, which lacks a pre-specified superclass structure”. However, in these cases, each group could be interpreted as containing only one class, so the methods could still be applied, if I understand the setup correctly. It would make sense to me to include these results and to compare MA-CS and MS-CS on these datasets.

---

> ### Author Rebuttal · Authors · 2026-03-30
>
> We thank the reviewer for their positive review. We address their concerns below.
>
> **W1, Q1 (Simplification of Theorem 4.5):**
>
> Your intuition is aligned with our discussion above Def. 4.4.
> Per sample, the prediction set size decreases if a decrease in out-of-group labels is "bigger" than an increase in in-group labels. Yet, it is not straightforward to establish a rigorous result on the expected size without assumptions on the score distribution and on the calibrated threshold.
> Assumps. 1-3 are reasonable for making the analysis tractable.
>
> Assumps. 2 and 3 are quite mild, while they assume continuous CDF of the scores they are not restricted to a specific distribution.
> Assump. 1 is also reasonable for the analysis. Despite reflecting an infinite calibration set, as stated, it allows "sparing cumbersome analysis of the effect of finite calibration sets on inclusion of a label in the predictions sets."
> Also, for theory of CP beyond marginal coverage, it is quite common to consider asymptotic calibration set size and there exist also works with much stronger assumptions, such as Bayes optimal classifiers [R1].
>
> [R1] Angelopoulos, Foygel Barber & Bates, "Theoretical foundations of conformal prediction," arXiv 2024.
>
> **W2, Q2 (Theorem 4.5 scope w.r.t. $\lambda$):**
>
> Theorem 4.5 characterizes $\mathrm{sign} ( \frac{d}{d\lambda} \mathbb{E}[ | \\mathcal{C}\_{\\lambda} (X) | ] |_{\lambda=0}  )$.
> Hence, it is a local result for $\lambda$ around 0. This will be further clarified.
> For practical setups ($p_1 \bar{n}_0 \ll p_0 \bar{n}_1$) it implies that there exists sufficiently small $\lambda>0$ such that the average prediction set size decreases due to our penalty.
> Importantly, our result does not require specific assumptions on the data and score distributions. Not being "context-dependent" can be seen as an advantage of our analysis.
>
> Our experiments verify that, indeed, there exist values of $\lambda>0$ that decrease the average prediction set size consistently for different scores in benchmark setups.
> The fact that in practice the range of these values is not small (see Figure 2, left) is a strength of our penalty method and is aligned with the intuition above Def. 4.4.
> While we would have been happy to extend the coverage of the theory, it is very common that a rigorous theoretical result does not capture the entire range of cases where a phenomenon occurs.
>
> **W3 (Theory for the model-specific variant):**
>
> Please note that the validity of our theory for the binary penalty does not depend on the class grouping quality. Indeed, it implies also a bad effect on the average set size if $p_1 \bar{n}_0 \gg p_0 \bar{n}_1$ (not common in practice).
> Thus, even without taking into account where the grouping comes from, we believe that replacing the binary penalty with continuous penalty significantly complicates the analysis, and we did not manage to generalizes our proof technique.
> Thus, we believe that it is fair to defer such extension for future research.
>
> **W4, Q3 (Explaining MA-CS vs. MS-CS performance differences):**
>
> First, note that both MA-CS and MS-CS use the same pretrained models, which underwent standard deep classification training, agnostic to the superclass structure.
>
> For the prediction set size metric, it is intuitive that the model-specific variant, MS-CS, yields smaller sets than the model-agnostic variant, MA-CS, because it utilizes similarity between classes as perceived by the model itself, while the predefined superclasses are less aligned with the model’s representation and can also be coarser.
> This is also aligned with Theorem 4.5 (which, as stated in the paper, motivates the model-specific extension): extracting class similarity directly from the model allows having smaller groups (larger $\bar{n}_1$, smaller $\bar{n}_0$) with higher in-group accuracy of the model (larger $p_0$, smaller $p_1$).
>
> For the #Superclasses metric, which is based on the predefined superclasses, we do see that in the majority of the cases MA-CS, which use this structure in its penalty, slightly outperforms MS-CS.
> The cases where the opposite holds, can be explained by the fact that MS-CS yields smaller prediction sets, which can naturally lead to including classes from a smaller number of superclasses.
>
> In the revision we will emphasize these explanations.
>
> **W5 (Notes on clarity and notation):**
>
> All the minor comments will be handled in the revision.
> Let us elaborate on the last comment.
>
> * Setting each group to be a singleton class is valid, and in fact, this is the variant with $M=I$ mentioned in the ablation paragraph below Figure 3, extensively examined in Appendix D.2, where it is named "MA-Diag".
> As stated in the paper: "The experiments in Appendix D.2 demonstrate the superiority of both MA-CS and MS-CS over this naive variant,..."
> As for AIR with singleton groups, note that it reduces to APS, known to have much larger prediction sets than (baseline) RAPS, that our methods consistently outperform.

---

> > ### Author Rebuttal · Reviewer_rkJU · 2026-04-01
> >
> > Thank you for the helpful response and addressing my concerns W1-W4-W5. Regarding W2-W3, I still think the theory is somewhat limited in scope (Thm 4.5 is local at $\lambda=0$, no theory for MS-CS) compared to what is shown in the experiments, but in this case I agree with the authors that it is fair to defer such theoretical extensions to future work. I see sufficient novelty to increase my score to 5, as long as the theoretical scope/limitations are indeed made more explicit in the revision.

---

> > > ### Author Response · Authors · 2026-04-01
> > >
> > > Dear Reviewer rkJU,
> > > we thank you again for your positive and thorough review.
> > > We will, of course, follow your request and make the theoretical scope and limitations more explicit in the revision.
> > > We are pleased that you stated that you see sufficient novelty in our work to increase your score to 5, and we would deeply appreciate it.

---

### Official Review · Reviewer_YmFf · 2026-03-16

**Soundness:** 2
**Presentation:** 2
**Significance:** 2
**Originality:** 2
**Overall Recommendation:** 4
**Confidence:** 4

**Summary:**

This paper proposes an augmented nonconformity score function for classification. It considers settings where labels exhibits grouping structure in the data and augment the nonconformity scores by adding a penalty term penalizing candidate labels that are semantically far from the top1 prediction. Depending on whether group annotations are available, the paper studies two approaches: a model-agnostic version based on human annotated groups, and a model specific version that uses a similarity matrix derived from the model. The proposed methods are evaluated on several image classification datasets.

**Compliance With Llm Reviewing Policy:**

Affirmed.

**Final Justification:**

My main concerns have been adequately addressed in the authors' rebuttal.

**Key Questions For Authors:**

1. Could the authors elaborate more on why MA-CS/MS-CS improve over the clustered method? In particular, what is the underlying mechanism that leads to the this observed gains?
2. The numerical results would be more convincing if the reliability metrics (marginal coverage and conditional coverage) were moved to the main body rather than deferred to the appendix. Since reliability is just as important as efficiency for conformal prediction methods. Conditional coverage may be especially informative and interesting to see here.
3. How does the accuracy of the pre-trained classifier affect the resulting conformal prediction set? The penalty term appears to rely implicitly on the top-1 predicted label being reasonably accurate, since it penalizes labels that deviate from that prediction. What would happen if the pre-trained classifier is weak or has low accuracy?

**Limitations:**

yes

**Strengths And Weaknesses:**

Strengths:
S1: The idea of introducing out-of-group penalty terms into the nonconformity scores to make the scores more adaptive to grouping structure is novel and interesting.
S2: The paper is overall well-structured and easy to follow.

Weaknesses:
W1: The experimental evaluation is somewhat limited. Current experiments focus only on image classification datasets with a single experimental setting. To better assess the efficiency and robustness of the proposed method, it might worth considering additional experimental configurations, such as varying the amount of calibration data, which is especially important for understanding behavior in small sample regimes given that the method requires an extra data split to tune $\lambda$. It might also be useful to study performance under different numbers of superclasses and potentially under distribution shift at test time.
W2: Although the paper discusses Hierarchical Conformal Classification (den Hengst et al., 2025) in the related work, it does not include a direct empirical comparison against this. Since the Hierarchical method explicitly accounts for label hierarchy when constructing conformal classification sets, it seems very relevant to the problem setting considered here. The paper would be strengthened either by including this benchmark or by providing a clear justification for why such a comparison is not appropriate.
W3: The empirical evidence for improvement is not entirely convincing. Although the authors report improvements on average, the error bars of the proposed method frequently overlap with some of the existing methods. This suggests that some of the reported improvements may not be statistically significant.

---

> ### Author Rebuttal · Authors · 2026-03-30
>
> We thank the reviewer for their review.
> We address their concerns below and hope that they will reconsider their score.
>
>
> **W1 (Regarding experimental coverage):**
>
> Our experiments cover multiple datasets (CIFAR-100, ImageNet, Mini-ImageNet, Living-17) and architectures (ResNet-34, ResNet-50, ViT), and 4 metrics (Size and #Superclasses in the main paper, and marginal and conditional coverage in the appendix)
> across 3 different scores with 5 variants per score, and for two typical values of $\alpha$.
> Thus, they provide a broad empirical basis.
>
> Following the comment we conducted additional experiments.
>
> The results in [**link**](https://postimg.cc/dkKHjhvK) reflect a small calibration regime (10% calibration / 90% test), where our methods consistently outperform the standard baseline. For example, on CIFAR-100 with ResNet-34 (RAPS), the average set size decreases from 5.77 to 4.52 (MA-CS) and 4.18 (MS-CS), while #Superclasses drops from 3.39 to $\sim$ 2.28--2.36.
>
> We also evaluate robustness under distribution shifts in [**link**](https://postimg.cc/t7vdXscS) and in [**link**](https://postimg.cc/zLD3bjbK) (Gaussian noise and brightness, following (Hendrycks et al., 2019)). Under brightness with CIFAR-100, ResNet-50, set size improves from 3.36 to 2.92 (MA-CS) and 2.74 (MS-CS), with #Superclasses reduced from 2.37 to $\sim$ 1.84 (RAPS). Similar gains hold under Gaussian noise.
>
> Overall, our methods remain effective across datasets, architectures, smaller calibration sets, and mild shifts. These and more results, which do not appear in the links due to limited response time, will be added to the revision.
>
> **W2 (Comparison with Hierarchical Conformal Classification):**
>
> We agree that the recent HCC (den Hengst et al., arXiv 2025) is a related work (and hence cited).
> HCC predicts hierarchical nodes while the leaves node are the classes. Thus, it is conceptually similar to (Goren et al., 2024), which we improved to better fit the two-level superclass structure of the examined datasets --- this is AIR that we already compare against.
> HCC also lacks publicly available code and does not outperform the standard baseline in efficiency (#covered leaves) (e.g., 4.3 vs. 2.67 in their table), while our approach already shows consistent improvement in efficiency.
> Thus, due to the above, we believe that it is acceptable not to include HCC in the experiments.
>
> **W3 (Consistent improvement by our methods even when error bars overlap):**
>
> We agree that overlapping error bars can obscure statistical significance when considering only aggregated results.
> Let us show that even in cases where there is overlap between the error bars, our methods consistently outperforms the baseline.
> To clarify these, we present per-trial comparisons over 100 runs for LAC in [**link**](https://postimg.cc/T566pPBX) and RAPS in [**link**](https://postimg.cc/23sq7FN4) for overlapping cases for CIFAR100-ResNet50 with $\alpha=0.05$. These show that improvements are consistent rather than driven by outliers. For LAC, MA-CS outperforms the baseline in 91/100 trials and MS-CS in 98/100; for RAPS, MA-CS does so in 93/100 trials and MS-CS in 100/100 trials.
>
> Thus, despite occasional overlap in error bars, the improvements are systematic and robust. We will include this analysis in the revised version.
>
> **Q1 (Improvement over clustered method):**
>
> The clustered method (Ding et al., 2023) aims to improve class-conditional coverage by calibrating a threshold per cluster, but it often does not outperform the standard baseline in efficiency (prediction set size).
> In contrast, MA-CS and MS-CS consistently outperform the baseline efficiency. They use a single global threshold based on a structured penalty that discourages including dissimilar classes, leading to smaller and more focused prediction sets (also supported by our theory). We believe this difference --- cluster-specific thresholds vs. global threshold and similarity-aware penalization --- explains the observed gains.
>
> **Q2 (Moving numerical results of coverage metrics):**
>
> We agree that reliability metrics, such as marginal and conditional coverage, are essential for evaluating conformal prediction methods alongside efficiency. Thus, we examined them thoroughly in the appendix (and discussed them in the paper). Using the additional allowed page, we will move numerical results to the main body in the revision.
>
> **Q3 (Effect of classifier's accuracy):**
>
> The classifier's accuracy affects the efficiency (set size) of CP in general.
> It should be noted that our penalty relies on group-wise correctness, which can be understood as a relaxation of top-1 correctness.
> Even though in all the examined benchmarks (with rather standard classifiers) our approach consistently enhanced CP, we agree with the reviewer that if the model is too weak then the gains may diminish, and we will state this as a limitation in the revision.

---

> > ### Author Rebuttal · Reviewer_YmFf · 2026-04-04
> >
> > I appreciate the authors’ thorough and detailed responses, as well as the additional experiments they provided. My main concerns have been adequately addressed, and I have adjusted my score upward accordingly.

---

> > > ### Author Response · Authors · 2026-04-05
> > >
> > > Dear Reviewer YmFf, we thank you again for your thoughtful and constructive review. We are pleased that you are satisfied with our response and sincerely appreciate the updated score.

---

### Official Review · Reviewer_cEN6 · 2026-03-18

**Soundness:** 2
**Presentation:** 2
**Significance:** 2
**Originality:** 2
**Overall Recommendation:** 3
**Confidence:** 5

**Summary:**

This paper proposes a method for constructing prediction sets with semantically coherent labels. To achieve this, the authors propose an out-of-group penalty that penalizes semantically far classes. Under some assumptions, they also theoretically show that the proposed penalty can reduce the average set size for any score function. The paper also includes a model-specific approach that does not assume knowledge to class structure. Finally, the authors demonstrate the performance of their proposed methods with baselines comparing the average set size and semantic coherence of sets.

**Compliance With Llm Reviewing Policy:**

Affirmed.

**Final Justification:**

I thank the authors for the detailed response and appreciate the inclusion of comparison with clustered CP. I would encourage the authors to report overall coverage metrics and not just the topCovGap for complete and fair comparison, following past work. I increase the score to 3, however I am still not in favor of acceptance as some of my original concerns are not fully addressed.

I still do not find the comparison between MA-CS vs. MS-CS rigorously explained, which makes the separation less clear and hence the motivation of the work (refer to my review and response below for detailed comments). The justification of assumptions was fairly limited in the original paper and not principled enough to establish formal understanding. I hope the authors consider this when they update the paper.

**Key Questions For Authors:**

1. The authors claim the importance of semantically coherent prediction sets, however what I found largely missing in the paper is discussion and formal comparison with conditional coverage guarantees. The paper includes a paragraph on this in the related work and claims clustered- and group-conditional methods result in larger prediction sets; however it does not comment on the utility of prediction sets. Ideally, we don’t just desire “prediction sets consisting of conditions that require similar treatment can help clarify appropriate next steps” (L29 in the paper), but also prediction sets that help in detecting surprising conditions or eliminating surprising but correct diagnoses by testing. Class-conditional coverage is one way to achieve this notion of validity. How do these different notions interact? Are there any tradeoffs, if so what? The paper makes a very strong claim regarding their target to begin with and does not include appropriate discussion on these points.

2. The model-specific variant uses class information, which is the same amount of information used by class-conditional and cluster-conditional methods. How do the methods differ in their goals and guarantees?

**Limitations:**

The authors have not discussed the limitations and potential negative societal impact of their work. The authors have included a brief conclusion section, however it does not discuss limitations of the presented method. I would encourage the authors to include an extended discussion of the limitations as well as expand their impact statement to discuss the potential negative societal impact of their work.

**Strengths And Weaknesses:**

**Soundness:** The authors have provided theoretical results to support their claims. While I found the theoretical results to be correct, I have a few concerns regarding the discussion and implications of results in the paper (some major and some minor regarding exposition):

- The discussion following Theorem 4.5 should be more principled with formal conditions to analyze the sign of the stated objective. L269 states “if labels are distributed uniformly, then the factor ... .This gives exact $a=b$.” However, there is no discussion of when the other conditions on $a$ and $b$ will hold, and how often? I did not find the statement “..teach us that there are many cases where sign $(a p_1 n_0 − b p_0 n_1) < 0$ even for non uniform conditional label distributions.” to be complete and appropriate to support the claim.
(Minor: authors should specify in the caption of Figure 1 that their result holds under certain assumptions e.g., small $\lambda$).

- Model-specific class similarity: The justification for the model-specific variant is incomplete currently in my view. While the notion of class-similarity seems desirable, how does this relate to the overall goal of the paper? Under what conditions on the model and data distribution will this variant achieve the desired goals? I ask this as there seems to be a gap in the goals being aimed for and the methods proposed to achieve this. This is also reflected in the empirical evaluation where the model-specific variant without additional information is able to outperform the model-agnostic variant (I comment on this in more detail below).
(Minor: the paper states in L310 “We believe that this is an interesting direction for future research, as establishing such a result may also help formally explain the benefits of certain score-specific regularization methods, such as the improved efficiency of RAPS compared to APS.” – the RAPS paper already has a result that proves their efficiency).


The empirical evaluation is extensive but several results lack sufficient explanation.

- The performance of model-specific and model-agnostic variants appear to have arbitrary patterns. For datasets with official semantic superclasses, MS outperforms MA in several places despite not using the superclasses. MA seems to be close to an oracle algorithm based on the paper’s narrative – it uses oracle superclass information and also has the desirable guarantees the paper wants to achieve. Despite this, the method falls short in practice. Moreover, MS consistently has smaller set sizes despite MA guaranteeing small set size (under certain assumptions). The paper attempts to justify this, however the explanation is not convincing in my view. Overall, this makes me question the effectiveness of the proposed MA algorithm as well as the relationship between the MA and MS variants.

- The discussion on comparison between the paper’s methods and AIR does not seem fair in my view. The paper claims AIR often achieves lower (i.e, better) values for the #Superclasses metric than their proposed methods while their methods achieve lower average set size. However, I do not believe this to be surprising given the $\lambda$ in their methods is tuned based on set size. I would request the authors to clarify this. If this is indeed the case, the comparison does not appear fair. This point in general needs to be addressed in the paper – which all methods are explicitly tuned for set size? Is it just the MA and MS methods proposed in the paper?


**Presentation:** The submission is overall structured well. However, gaps in several explanations and missing discussion hampers the understanding of the paper. I have pointed these gaps across several points in my review. Overall, these have resulted in difficulty in grasping the goals of the paper and the significance of the methods in achieving the goals. Minor comment regarding Figure 3: the labels are extremely difficult to read even after zooming in.

**Significance:** The paper aims to address the problem of constructing semantically coherent and on average smaller prediction sets. Broadly, the goal appears desirable; however, without appropriate discussion of the relationship of this target with other desirable targets e.g., conditional coverage (refer to questions below for detailed comments) and further justification of the performance of the methods, it is hard to ascertain the overall significance and utility.


**Originality:** The idea of using penalties in the conformal score function to achieve specific desiderata is common in the literature. The novelty of this paper lies in the penalty they propose that penalizes distance between classes based on group membership. I believe the proposal would be stronger if the approach (especially model-specific) is further justified and positioned in the context of prior and concurrent work.

---

> ### Author Rebuttal · Authors · 2026-03-30
>
> We thank the reviewer for their review. We address their concerns below and hope that they will reconsider their score.
>
> **W1 (The discussion below Theorem 4.5):**
>
> The discussion aims to justify why in typical practical setups:
> $a p_1 \bar{n}_0 - b p_0 \bar{n}_1 < 0$
> The key observation: the ratio $\frac{p_0 \bar{n}_1}{p_1 \bar{n}_0}$ is generally large, driven by two factors:
> * Cardinality ratio $\bar{n}_1 / \bar{n}_0$:
> Here, $\bar{n}_0$ (resp. $\bar{n}_1$) is the average number of in-group (resp. out-of-group) classes.
> For a dataset with $G$ groups of roughly equal size, this ratio scales as $G-1$. For example, in CIFAR100 it equals $19$.
> * Probability ratio $p_0 / p_1$:
> Here, $p_0$ denotes the model's group-level accuracy, and $p_1=1-p_0$. For a well-trained model, $p_0$ is typically high, implying that $p_0 / p_1$ is also large. In fact, top-1 accuracy above 0.5 already ensures that this ratio is larger than 1.
>
> Hence, $\frac{p_0 \bar{n}_1}{p_1 \bar{n}_0}$ tends to be large in practice. While the implication $<0$ is most immediate when $a \approx b$ (e.g., under a uniform conditional class distribution), the magnitude of $\frac{p_0 \bar{n}_1}{p_1 \bar{n}_0}$ ensures that there exist a wide range of $a,b$ for which the inequality still holds.
> This will be emphasized in the revision.
>
> **W2 (Relevance of the model-specific variant to the paper's goal):**
>
> The overall goal of the paper is to enhance CP using class similarity, as stated in the title.
> A key message is that class similarity can improve CP efficiency (prediction set size) when it follows predetermined human-based semantics, and—even better—when it comes from how the model perceives the data.
>
> The paper's organization reflects our path toward enhancing CP via class similarity.
> First, we present a model-agnostic (MA) method that assumes a given partition of classes into semantic groups.
> Having coherent classes in the prediction set can facilitate interpretability and downstream actions.
> Yet, a major goal of the paper, which as discussed distinguishes it from earlier works that use class structure, is not compromising on efficiency (set size).
> Hence, the design of our MA method follows the goal of having prediction sets that are both small and contain a limited number of distinct groups.
> Next, we theoretically analyze the method and reveal the factors that actually improve the baseline's efficiency: partition to small groups aligned with the model's in-group accuracy.
> This directly motivates the design of our model-specific (MS) variant, which extracts class similarity directly from the model --- focusing on the goal of reducing the prediction set size.
> The MS variant also eliminates the need for a manually defined semantic partition, which may not always be available.
>
> We will further emphasize the above in the revision.
>
> **W3 (Explaining MA-CS vs. MS-CS  performance differences):**
>
> Due to space limitation, we refer you to our response to **Reviewer rkJU, W4** that addresses this comment.
>
> **W4 (The comparison with AIR):**
>
> The comparison with AIR is appropriate, as it highlights two fundamentally different paradigms: enforcing semantic coherence as a hard constraint (AIR) versus incorporating it as a soft, regularized objective (MA/MS-CS). The fact that our methods achieve significantly smaller prediction sets while remaining competitive in terms of #Superclasses is a central result, which is also supported by our theory.
>
> Note also that for RAPS and SAPS baselines the hyperparams are already tuned for set size (see Appendix B). Yet, our approach still consistently improves their efficiency.
>
> **Q1 (Conditional coverage):**
>
> We agree that conditional coverage is an important aspect in CP. Thus, we already consider a worst-class-conditional metric in the paper.
>
> As stated in page 7, line 380:
> "In Appendix C.2, we report the worst class-conditional coverage gap for each method across all settings. Overall, our class-similarity approach does not significantly change this metric compared to the standard versions. In fact, for LAC and SAPS, our MS-CS often yields modest improvement."
>
> Also stated: "In Figure 2(right), we present the effect of $\lambda$ on the worst class-conditional coverage gap. As shown, this metric remains relatively stable across the range of $\lambda$ values." That is, there is no tradeoff.
>
> **Q2 (class/cluster-conditional methods):**
>
> Class/cluster-conditional methods aim to guarantee coverage to each group (e.g., Mondrian CP applies calibration per group). They often require many calibration samples per group and produce large prediction sets.
> In contrast, MS/MA-CS focus on building smaller prediction sets with fewer superclasses. Empirically, our methods maintain stable class coverage while achieving state-of-the-art efficiency.
>
> **Other comments:**
>
> All the minor comments will be handled in the revision, including expanding the discussion on the limitations and societal impact using the additional allowed page.

---

> > ### Author Rebuttal · Reviewer_cEN6 · 2026-04-03
> >
> > I thank the authors for their responses. Unfortunately, most of my concerns still remain as I detail below. That said, I will consider the authors' follow up response in the final justification.
> >
> > > Theorem 4.5 discussion
> >
> > The authors' explanation still is not principled enough to establish formal understanding. What if the groups are not of equal size? (this can certainly be true in practice e.g., long-tailed datasets, which the authors do not consider in their evaluation). What if the model is not trained well? Such properties should be a part of the discussion and show how the conditions weaken in such cases.
> >
> > > "Yet, a major goal of the paper, which as discussed distinguishes it from earlier works that use class structure, is not compromising on efficiency (set size)."
> >
> > I understand that but prior works provide class-conditional guarantees which this method doesn't. My key question (1) is still largely unresolved. Authors state in the rebuttal "They often require many calibration samples per group and produce large prediction sets." -- this is true for Mondrian CP but not cluster CP, which was introduced precisely to alleviate this. In addition, clusterCP also performs well in comparison to baselines on long-tailed datasets, which this paper does not discuss or evaluate on. I am not establishing the latter is strictly better than the proposal, but the lack of comparison and discussion is concerning in my view. Moreover, such tradeoffs are not acknowledged in the paper.
> >
> > > MA-CS vs. MS-CS
> >
> > I believe the justification for this point is not sufficient and lacks rigor. The authors state in their rebuttal "For the #Superclasses metric, which is based on the predefined superclasses, we do see that in the majority of the cases MA-CS, which use this structure in its penalty, slightly outperforms MS-CS.", however I do not see a consistent pattern here. There are some cases e.g., in Table 1 where it does not outperform MS-CS. This is concerning as it brings me back to the question I raised, when should one even use MA-CS? If the paper was motivated from this method, at least in the oracle cases this method should clearly outperform MS-CS? Otherwise, why not just establish MS-CS as the goal in the first place. More generally, performance patterns are arbitrary, and guidance on using one method over the other and principled justifications are largely missing.

---

> > > ### Author Response · Authors · 2026-04-05
> > >
> > > Dear reviewer cEN6,
> > > we thank you for this follow-up and address your comments below. We hope that, given this and our previous responses, you will reconsider your score.
> > >
> > > **More on the discussion below Theorem 4.5:**
> > >
> > > As discussed in our previous response (W1), the key observation is that the penalty's efficiency gain is implied when the ratio $\frac{p_0 \bar{n}_1}{p_1 \bar{n}_0}$ is large, which is the case in the benchmark setups.
> > > Even when groups are of different sizes, $\bar{n}_1 / \bar{n}_0$ (average out-of-group/in-group cardinality ratio) is expected to be large in typical setups.
> > > In fact, unless a single group contains more than half of the classes it is already ensured that this ratio is above 1.
> > > As for the model quality, $p_0/p_1$ (the model's group-level accuracy/error ratio) is also expected to be larger than 1 for reasonable models.
> > > In fact, as mentioned in our previous response, top-1 accuracy above 0.5 already ensures that this ratio is larger than 1.
> > >
> > > This discussion, which include formal examples of very weak conditions under which the ratio is larger than 1, explains why our theory supports the improved efficiency due to the proposed penalty that is consistently observed in our broad set of experiments.
> > >
> > > We will use the additional allowed page in the final version to extend and clarify the discussion.
> > > We will also emphasize there that, theoretically, there exist extreme cases (extremely dominant groups and very weak models) where the ratio may be lower than 1. Notably, Theorem 4.5 is valid also in this regime and implies a bad effect on the average set size.
> > > Yet, such extreme cases have not been encountered in none of the many benchmark setups that we examined.
> > >
> > > **Our approach consistently outperforms Clustered CP in efficiency and is competitive in class-conditional coverage over broad benchmark setups:**
> > >
> > > First, please note that in page 6, line 300, we already state that the Clustered CP "extends and improves the efficiency of class-wise Mondrian CP..."
> > > We will discuss it also in the related work section.
> > > Yet, both our experiments and those in the original (Ding et al., 2023)
> > > show that Clustered CP typically does not outperform the standard baseline in terms of average
> > > prediction set size, which is a central metric in our work and in the CP literature in general.
> > >
> > > Next, as we previously pointed in our response to the reviewer's Q1: 1) our paper already includes a comparison of a worst-class-conditional coverage metric across the methods and setups (Appendix C.2), which is discussed in the main paper; and 2) in our approach, there is no trade-off pattern between this metric and the others metrics (Figure 2(right) and Tables 5 \& 6).
> > >
> > > Importantly, Tables 1 \& 2 show that
> > > both variants of our approach **consistently outperform Clustered CP in terms of set size**.
> > > Tables 5 \& 6 show that our approach is **competitive with Clustered CP in terms of the conditional coverage metric** (note that the class-conditional guarantees of Clustered CP have practical limitations --- see their Prop. 3).
> > > Overall, our experiments show clear advantages of our approach over Clustered CP in a broad set of benchmark setups.
> > >
> > > Despite already having a broad experimental coverage, following the reviewer's comment we conducted additional experiments using CIFAR-100 with a long-tailed test distribution. The model is trained on the balanced training set, while we subsample the CIFAR-100 test set to follow a long-tailed distribution with imbalance ratio $\rho = 0.1$, and split it to 20\% calibration and 80\% test, as in the paper.
> > >
> > > Results for ResNet-50 and RAPS and LAC scores are provided in **https://postimg.cc/8sD7p3jk**.
> > > They show that our methods outperform Clustered CP in all metrics.
> > >
> > > **MA-CS vs. MS-CS:**
> > >
> > > Note that MA-CS, on its own, consistently outperforms all the baselines – Standard, Clustered, AIR – in terms of average prediction set size, while also outperforming Standard and Clustered in terms of \#Superclasses, and narrowing the gap with AIR, whose low \#Superclasses comes at the cost of very poor efficiency.
> > > The fact that our MS-CS variant, which adapts to the model’s representation, further improves the prediction set size strengthens our overall contribution, even if it may have lower \#Superclasses than MA-CS in some cases.
> > >
> > > **There are natural reasons to use MA-CS** when a reliable group structure is given: 1) it is accompanied by more established theoretical support (in particular for reduction in \#Superclasses compared to the standard baseline); and, importantly, 2) it is applicable for black-box models and does not require access to training data (unlike MS-CS, which utilizes the internal network's embeddings and training data to extract the model-specific class similarity).
> > > We agree that this distinction was not sufficiently emphasized and will clarify it in the final version.

---

### Decision · Program_Chairs · 2026-04-30

**Decision:**

Accept (regular)

**Comment:**

This paper augments CP score functions with a class-similarity penalty, proposing model-agnostic (MA-CS) and model-specific (MS-CS) variants. Reviewers agree that the core idea, i.e., augmenting CP scores with a class-similarity penalty, is well-motivated, and the experiments consistently show improvements over baselines. The disagreement centers on the paper's framing and theoretical scope. Final ratings: 6/5/4/3.

The main concern, raised by Reviewer cEN6 and partly echoed by Reviewer rkJU, is the relationship between MA-CS and MS-CS. The paper's narrative places MA-CS at the center with theoretical backing (Theorem 4.5), but MS-CS frequently outperforms it in set size without requiring predefined superclass information. As Reviewer rkJU notes, the theoretical guarantees are relative to the unpenalized baseline, so the results are not inconsistent with theory. Nevertheless, the paper does not adequately explain this empirical pattern or provide practical guidance on when to prefer one variant over the other.

A secondary concern is theoretical scope: Reviewer cEN6 views it as a significant weakness that Theorem 4.5 is local ($\lambda \to 0$) and covers only MA-CS. Reviewer rkJU acknowledges the gap but does not consider it grounds for rejection.

Overall, the contribution is solid and merits acceptance. The core idea is well-motivated and consistently supported by experiments. As a critical revision requirement, the authors are encouraged to provide a clear discussion of the connection between MA-CS and MS-CS, including principled guidance on when each variant is preferred, instead of leaving the frequent empirical superiority of MS-CS as an unexplained observation.